# RL Tango: Reinforcing Generator and Verifier Together for Language Reasoning

**Kaiwen Zha**[1,*]    **Zhengqi Gao**[1,*]    **Maohao Shen**[1]    **Zhang-Wei Hong**[2]
**Duane S. Boning**[1]    **Dina Katabi**[1]

[1]MIT        [2]MIT-IBM Watson AI Lab

## Abstract

Reinforcement learning (RL) has recently emerged as a compelling approach for enhancing the reasoning capabilities of large language models (LLMs), where an LLM generator serves as a policy guided by a verifier (reward model). However, current RL post-training methods for LLMs typically use verifiers that are fixed (rule-based or frozen pretrained) or trained discriminatively via supervised fine-tuning (SFT). Such designs are susceptible to reward hacking and generalize poorly beyond their training distributions. To overcome these limitations, we propose TANGO, a novel framework that uses RL to concurrently train both an LLM generator and a verifier in an interleaved manner. A central innovation of TANGO is its generative, process-level LLM verifier, which is trained via RL and co-evolves with the generator. Importantly, the verifier is trained solely based on outcome-level verification correctness rewards without requiring explicit process-level annotations. This generative RL-trained verifier exhibits improved robustness and superior generalization compared to deterministic or SFT-trained verifiers, fostering effective mutual reinforcement with the generator. Extensive experiments demonstrate that both components of TANGO achieve state-of-the-art results among 7B/8B-scale models: the generator attains best-in-class performance across five competition-level math benchmarks and four challenging out-of-domain reasoning tasks, while the verifier leads on the ProcessBench dataset. Remarkably, both components exhibit particularly substantial improvements on the most difficult mathematical reasoning problems. Code is at: https://github.com/kaiwenzha/rl-tango.

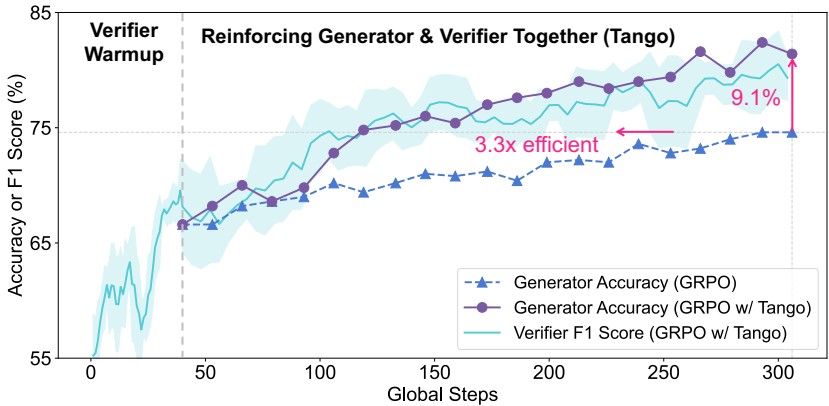

Figure 1: **Generator and verifier training dynamics of TANGO.** The generator and verifier co-evolve through mutual reinforcement. Supported by the verifier, the generator of TANGO achieves significantly better training efficiency and stronger final performance compared to vanilla GRPO. The generator accuracy is the pass@1 accuracy on the MATH 500 dataset, and the outcome F1 score of the verifier is reported on training data.

---

*Equal contribution.

39th Conference on Neural Information Processing Systems (NeurIPS 2025).

# 1 Introduction

Large language models (LLMs) have recently demonstrated remarkable capabilities across a broad spectrum of natural language processing (NLP) tasks [22, 5, 16]. Despite their impressive performance, pretrained LLMs often struggle with complex reasoning tasks requiring multi-step thinking and planning [23, 19]. To enhance reasoning abilities, LLMs typically undergo post-training via supervised fine-tuning (SFT) [61, 10] or reinforcement learning (RL) [23, 19]. SFT teaches models to mimic curated solutions, but this imitation-based training lacks interaction and generalizes poorly to unfamiliar reasoning paths [9]. In contrast, RL frames learning as an active exploration process, where models learn from experience and directly optimize for task success through trial and feedback, enabling stronger generalization [9]. Therefore, RL has become a central component of recent LLM post-training, with large-scale industrial deployments such as OpenAI's o1 [23] and DeepSeek R1 [19] demonstrating its effectiveness in unlocking advanced reasoning capabilities.

In LLM post-training with RL, the LLM generator acts as the policy model, where each action corresponds to generating the next token based on the current sequence (the state). A reward model, commonly known as a verifier, assesses the quality of the generated outputs and provides feedback that is used to guide the generator's policy updates using RL algorithms [41, 40, 43, 1].

However, a critical limitation of current RL post-training approaches [43, 50, 60] is their reliance on a fixed verifier, typically implemented using rules-based metrics or a frozen pre-trained reward model. This fixed verifier limits the potential improvement of the generator and is vulnerable to reward hacking in distribution changes [14]. Ideally, verifiers should be trained jointly with generators [30], enabling mutual improvement. Yet, designing an effective co-evolving system remains challenging. Among recent attempts, PRIME [12] is, to the best of our knowledge, the only approach that trains the generator alongside the verifier. However, PRIME's verifier still faces critical shortcomings. First, it employs a discriminative logit-based process reward model (PRM) that generates deterministic reward signals, making it susceptible to reward hacking [13], despite being trained online. Second, PRM is trained using SFT with outcome-level labels, despite these labels being collected in an online manner. SFT significantly restricts verifier reasoning capabilities and generalization potential [9].

We argue that the effectiveness of a co-evolving system critically relies on the capabilities of both the generator and the verifier. If one component is significantly weaker and lags behind, it can impede the overall learning dynamics and limit mutual improvement. An effective co-evolutionary framework requires both agents to be robust and to continuously enhance each other's performance. To this end, we introduce TANGO, *a novel framework that jointly trains an LLM generator and an LLM verifier in an interleaved manner via RL*. Unlike existing methods that use frozen, discriminative, or SFT-trained reward models [43, 60, 12], TANGO introduces a process-level, *generative* LLM verifier that is *trained via RL* and *co-evolves alongside the generator* throughout training. Specifically, the generator produces multi-step reasoning trajectories, while the verifier offers natural language feedback comprising both step-level assessments and an overall correctness judgment. The generator leverages gold outcome-level correctness signals combined with detailed step-level rewards from the verifier, improving the efficiency of policy learning [48], and guiding the generator toward more robust reasoning strategies [30]. Importantly, the verifier is trained exclusively using outcome-level verification correctness rewards, *without process-level annotations*. Through RL, it progressively refines its chain-of-thought [55] verification reasoning, gradually aligning its step-level feedback with final correctness outcomes as the generator's reasoning trajectories evolve.

TANGO offers a more effective design for the co-evolving generator-verifier system, addressing the limitations of previous approaches. First, by training the verifier using RL rather than SFT, the verifier develops stronger reasoning skills and generalizes better beyond supervised imitation. This mirrors the rationale behind preferring RL over SFT when training generators under outcome-only supervision. Second, the generative and sampling-based nature of TANGO's verifier introduces stochasticity into the reward signals, enhancing its robustness against reward hacking. Consequently, through interleaved training, the generator and verifier mutually reinforce each other, enabling improved reasoning strategies and superior generalization performance, as shown in Figure 1.

We conduct extensive experiments to evaluate the effectiveness of TANGO across diverse reasoning tasks and experimental settings. Compared to vanilla RL methods trained only on outcome-level rewards, TANGO achieves an average relative improvement of 25.5% on five competition-level math benchmarks and 7.3% on four challenging out-of-domain reasoning tasks, consistently across three RL algorithms. Remarkably, TANGO with GRPO doubles the accuracy on the most challenging

benchmark, AIME 2025, relative to vanilla GRPO. Furthermore, TANGO substantially outperforms ORM- and PRM-based baselines, including PRIME. In a comprehensive comparison with prior state-of-the-art LLM reasoning methods, TANGO establishes new state-of-the-art results among 7B/8B-scale models, delivering the best performance on the most difficult tasks, namely AIME 2025, AIME 2024, and AMC 2023. TANGO verifier also sets a new state-of-the-art on ProcessBench [64], despite not using process-level annotations. In particular, it achieves the highest step-level verification performance on the most challenging subsets, OlympiadBench and Omni-MATH, significantly surpassing previous methods, including the much larger Qwen2.5-Math-72B-Instruct model even though our verifier is initialized from just a Qwen2.5-7B base model. Finally, an in-depth analysis on an algorithmic reasoning task with available gold step-level labels confirms that TANGO effectively bootstraps both the generator and verifier into highly capable states through mutual reinforcement.

## 2 Related Work

**RL for LLM reasoning.** RL was initially used to align LLM outputs with human preferences, enhancing response quality, instruction-following, and style [8, 47, 67, 39, 25]. As LLMs expanded into domains demanding multi-step reasoning and structured problem-solving such as mathematics, coding [24, 2], and web navigation [66, 29], RL has evolved beyond basic alignment toward enhancing LLM reasoning abilities [23, 19, 43, 27, 12, 60, 44, 37, 52, 7, 53]. A pivotal milestone was OpenAI's o1 [37, 23], which demonstrated RL's effectiveness at scale. Subsequent research has further advanced RL-based LLM reasoning through improved optimization algorithms [41, 40, 43, 1], curriculum-based training [46, 60], and enriched evaluation benchmarks [11, 64, 49].

**Reward modeling.** Reward signals are critical for guiding LLM post-training toward desirable behaviors and enabling effective inference-time scaling. Reward models (RMs a.k.a. verifiers) are categorized as outcome reward models (ORMs) or process reward models (PRMs) based on the granularity of evaluation. ORMs [39, 19, 35] assign a single scalar reward to the final token in a response trajectory, resulting in sparse supervision. Typically discriminative, ORMs attach a classification head to pretrained LLMs and are trained via SFT on labeled response pairs (preferred vs. rejected) [67, 47, 12, 32]. Recently, generative ORMs have emerged [62, 33], producing explicit rationales before assigning outcome scores.

In contrast, PRMs [12, 7, 31, 53, 32, 54] provide fine-grained, step-level feedback throughout the generation trajectory, facilitating more precise credit assignment [30] and improved training efficiency [48]. This detailed feedback helps models efficiently explore policy spaces and develop stronger reasoning capabilities. However, most existing PRMs remain discriminative, frozen, and deterministic, rendering them brittle to distribution shifts and reward hacking [42], while also requiring expensive step-level annotations. PRIME [12] partially addresses these issues by jointly training a PRM via SFT alongside the generator, reducing annotation overhead. Yet, PRIME's logit-based, deterministic rewards still leave it vulnerable to hacking, and its SFT-based training constrains generalization. To address these limitations, we propose a generative, process-level verifier (generative PRM) that outputs stochastic, step-level rewards as textual judgments (e.g., "Correct" or "Incorrect"). Unlike existing PRMs, both logits-based and generative, that rely exclusively on SFT, ours is the first PRM trained using RL. We note that the idea of generative PRMs traces back to LLM-as-a-judge [65], which uses frozen LLMs for scoring. More recent works [28, 63] have explored SFT-based generative PRMs for inference time scaling, but these concurrent approaches remain orthogonal to our work, as none involve RL-trained, co-evolving generative PRMs.

## 3 Method

### 3.1 Preliminaries

We denote the autoregressive LLM generator and verifier as $\pi_g$ and $\pi_v$, respectively. For notational simplicity, we use $\pi_\theta$ as a unified symbol when an equation applies to both models. RL aims to optimize a policy model by maximizing the expected cumulative discounted reward through interactions with the environment, i.e., by taking actions and transitioning between states. In the context of RL post-training for LLMs, the policy model corresponds to the LLM generator or verifier. The state at step $t$ is defined as the combination of the input prompt (i.e., a question) $\mathbf{x}$ and the partially generated response $\mathbf{o}_{<t}$, while the action corresponds to generating the next token $o_t$. To

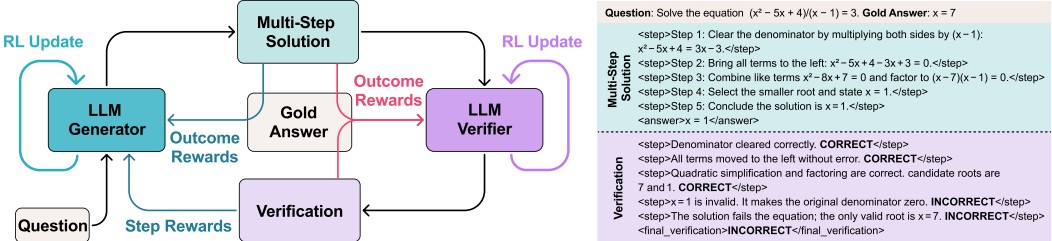

Figure 2: **Overview of the TANGO framework with a generation and verification example.** Given a question, the LLM generator produces a multi-step solution, which is then evaluated by the LLM verifier. The generator is trained using both step-level rewards from the verifier and outcome-level rewards based on its final answer, while the verifier is trained only with outcome-level rewards based on the correctness of its final judgment and format.

optimize $\pi_\theta$, policy gradient methods are employed to estimate the gradient of the expected reward with respect to the policy parameters $\theta$. Built upon it, a widely used surrogate objective [41, 40, 43, 1] is formulated using importance sampling:

$$\mathcal{J}(\theta) = \mathbb{E}_{\substack{(\mathbf{x}, \mathbf{y}) \sim \mathcal{P} \\ \mathbf{o} \sim \pi_{\theta_{\text{old}}}(\cdot | \mathbf{x})}} \left\{ \frac{1}{|\mathbf{o}|} \sum_{t=1}^{|\mathbf{o}|} \left[ \min \left( c_t(\theta) \hat{A}_t, \, \text{clip}\left( c_t(\theta), 1 - \epsilon, \, 1 + \epsilon \right) \hat{A}_t \right) - \beta \, \mathbb{D}_{\text{KL}} \left[ \pi_\theta \, \| \, \pi_{\theta_{\text{ref}}} \right] \right] \right\}, \tag{1}$$

$$c_t(\theta) = \frac{\pi_\theta(o_t \mid \mathbf{x}, \mathbf{o}_{<t})}{\pi_{\theta_{\text{old}}}(o_t \mid \mathbf{x}, \mathbf{o}_{<t})},$$

where $\pi_\theta$, $\pi_{\theta_{\text{old}}}$, and $\pi_{\theta_{\text{ref}}}$ denote the current, old, and reference policy models respectively, $|\mathbf{o}|$ is the sequence length, $\hat{A}_t$ is an estimator of the advantage at step $t$, and $\mathbf{y}$ is the gold answer used to compute the reward and subsequently the advantage $\hat{A}_t$. The hyperparameter $\epsilon$ controls the clipping range of the importance sampling ratio, while $\beta$ regulates the KL-divergence penalty strength. RL algorithms mainly differ in their methods for estimating $\hat{A}_t$, such as group-normalized rewards (GRPO [43]), leave-one-out reward averaging (RLOO [1]), or batch-normalized rewards (REINFORCE++ [21]).

## 3.2 TANGO

TANGO jointly trains an LLM generator and an LLM verifier via interleaved RL, creating a self-reinforcing loop where each agent iteratively strengthens the other. Figure 2 illustrates the overall TANGO framework. Specifically, we alternate training the generator policy $\pi_g$ for $N_g$ steps and the verifier policy $\pi_v$ for $N_v$ steps, repeating this cycle iteratively. Below, we detail the RL training for each component. Please refer to Appendix C for the detailed algorithm flow of TANGO, and Appendix F for more generation and verification examples.

**RL-based LLM generator.** Given a question-answer pair $(\mathbf{x}, \mathbf{y})$ from the training distribution $\mathcal{P}$, the generator $\pi_{g_{\text{old}}}$ produces a step-by-step solution $\mathbf{o}_g \sim \pi_{g_{\text{old}}}(\cdot \mid \mathbf{x})$. Our reward design is as follows:

- **Rule-based outcome-level rewards**: Extract the predicted answer $\hat{\mathbf{y}}$ from the generated solution $\mathbf{o}_g$, and compute an analytical rule-based outcome-level correctness reward:

$$r_{g,\text{out}}(\mathbf{o}_g) = \begin{cases} 1, & \text{if } \hat{\mathbf{y}} = \mathbf{y}, \\ 0, & \text{otherwise}. \end{cases} \tag{2}$$

- **Step-level rewards from the verifier**: We prepare a verification prompt using the question $\mathbf{x}$ and the generator solution $\mathbf{o}_g$, and then sample a verification response from the verifier $\mathbf{o}_v \sim \pi_v(\cdot \mid \mathbf{x}, \mathbf{o}_g)$. If there are $K$ reasoning steps in the generator response $\mathbf{o}_g$, then the response $\mathbf{o}_v$ will also contain $K$ step-wise judgments $y_{\text{step},k} \in \{-1, 1\}$, where $k = 1, 2, \ldots, K$, with $-1$ denoting 'Incorrect' and $1$ denoting 'Correct'. Please refer to the right part of Figure 2 for an example on $\{\mathbf{x}, \mathbf{y}, \mathbf{o}_g, \mathbf{o}_v\}$. Finally, the step-level rewards are computed as:

$$\mathbf{R}_{g,\text{step}} = \left\{ r_{g,\text{step}}^{I(1)}(\mathbf{o}_g), \ldots, r_{g,\text{step}}^{I(K)}(\mathbf{o}_g) \right\}, \quad r_{g,\text{step}}^{I(k)}(\mathbf{o}_g) = \frac{y_{\text{step},k}}{K} \in \left\{ \frac{-1}{K}, \frac{1}{K} \right\}, \tag{3}$$

where $I(k)$ is the index of the end token in the generator's $k$-th reasoning step ($k = 1, 2, \ldots, K$). We normalize the reward by the number of reasoning steps to remove policy's bias toward step length, allowing the generator to adaptively determine an appropriate number of steps based on the problem. Essentially, our approach adopts a generative process-level verifier that produces natural language judgments, enabling stochastic sampling-based step-wise evaluations.

We compute advantages separately for outcome-level and step-level rewards, combining them through a weighted sum. Using GRPO [43] as an illustrative example (though TANGO is compatible with other RL algorithms [1, 41], as shown in Section 4), the generator policy $\pi_{g_{\text{old}}}$ samples a group of $M$ responses $\{\mathbf{o}_g^i\}_{i=1}^M$, which are evaluated by the verifier to produce verification outputs $\{\mathbf{o}_v^i\}_{i=1}^M$. Each data sample $(\mathbf{o}_g^i, \mathbf{o}_v^i)$ contains $K^i$ reasoning and verification steps. Below are the advantages:

- **Outcome-level advantages:** The outcome-level advantage of the $i$-th response $\hat{A}_{g,\text{out},t}^i$ is calculated by normalizing the group-level outcome rewards $\mathbf{R}_{g,\text{out}} = \{r_{g,\text{out}}(\mathbf{o}_g^i)\}_{i=1}^M$:

$$\hat{A}_{g,\text{out},t}^i = \frac{r_{g,\text{out}}(\mathbf{o}_g^i) - \text{mean}(\mathbf{R}_{g,\text{out}})}{\text{std}(\mathbf{R}_{g,\text{out}})}. \tag{4}$$

- **Step-level advantages:** For step-level advantages, group-level normalization is performed across all step reward elements from all responses within the group, i.e.,

$$\mathbf{R}_{g,\text{step}} = \bigcup_{i=1}^M \mathbf{R}_{g,\text{step}}^i = \bigcup_{i=1}^M \left\{ r_{g,\text{step}}^{I(1)}(\mathbf{o}_g^i), \ldots, r_{g,\text{step}}^{I(K^i)}(\mathbf{o}_g^i) \right\}. \tag{5}$$

To clarify, the set size $|\mathbf{R}_{g,\text{step}}| = \sum_{i=1}^M K^i$. Next, the step advantage of each token $\hat{A}_{g,\text{step},t}^i$ is calculated as the sum of the normalized rewards of its following steps:

$$\hat{A}_{g,\text{step},t}^i = \sum_{\{k|I(k) \geq t\}} \frac{r_{g,\text{step}}^{I(k)}(\mathbf{o}_g^i) - \text{mean}(\mathbf{R}_{g,\text{step}})}{\text{std}(\mathbf{R}_{g,\text{step}})}. \tag{6}$$

We note that for a given sample (i.e., with fixed index $i$), the outcome-based advantages $\hat{A}_{g,\text{out},t}^i$ are the same across all tokens indexed by $t$, whereas the step-level advantages $\hat{A}_{g,\text{step},t}^i$ may vary across tokens. The final advantage is derived by blending the outcome advantage Eq. (4) and the step advantage Eq. (6) with a hyperparameter $\alpha \in (0, 1)$:

$$\hat{A}_{g,t}^i = (1 - \alpha)\hat{A}_{g,\text{out},t}^i + \alpha\hat{A}_{g,\text{step},t}^i. \tag{7}$$

We highlight two key design choices that are crucial to the success of our generator training:

- We apply an exponential decay schedule to $\alpha$, which is essential to TANGO's success. Early in training, step-level supervision has a stronger influence to encourage exploration of reasoning strategies. As training progresses, we gradually reduce its weight to promote stable convergence and mitigate reward hacking.

- Empirically, we find that computing and normalizing the step and outcome advantages separately before combining them yields significantly more stable learning than merging the rewards first and computing a single advantage. This is because advantage normalization depends on the scale and distribution of the underlying rewards. Merging step and outcome rewards before normalization could distort their relative contributions due to scale mismatch, resulting in instability and degraded performance. By normalizing each advantage independently, we preserve their intended effects prior to aggregation.

**RL-based LLM verifier.** The verifier generates a verification response $\mathbf{o}_v$ conditioned on the question $\mathbf{x}$ and the generator's solution $\mathbf{o}_g$. As the example shown in Figure 2, the verifier's final judgment label $y_{\text{final}} \in \{0, 1\}$ can be extracted from $\mathbf{o}_v$, where $y_{\text{final}} = 1$ indicates the verifier considers the generator's solution correct, and $y_{\text{final}} = 0$ indicates incorrect. Given that the correctness of the generator's answer is known from Eq. (2), we define the verifier's outcome-level reward based on how well its final judgment matches this ground-truth correctness, as well as its format score:

$$r_{v,\text{out}}(\mathbf{o}_v) = r_{v,\text{correct}}(\mathbf{o}_v) + \gamma \cdot r_{v,\text{format}}(\mathbf{o}_v), \quad r_{v,\text{correct}}(\mathbf{o}_v) = \begin{cases} 1, & \text{if } y_{\text{final}} = r_{g,\text{out}}(\mathbf{o}_g), \\ 0, & \text{otherwise.} \end{cases} \tag{8}$$

The default value of $r_{v,\text{format}}(\mathbf{o}_v)$ is set to 1.0 and is gradually reduced for each unmet formatting criterion, such as the absence of step-wise justifications or discrepancies in step numbering between the verifier output $\mathbf{o}_v$ and the generator output $\mathbf{o}_g$. The hyperparameter $\gamma \in \mathbb{R}^+$ controls the contribution of the format score in the final outcome-level reward. The verifier is trained without any process-level supervision, eliminating the need for step-level annotations. Empirically, we observe that although the verifier is trained solely with outcome-level signals, it progressively learns to produce accurate step-level judgments by refining its chain-of-thought verification reasoning through RL training, thereby providing useful guidance to the generator.

A key challenge we observe in training the verifier is class imbalance in early stages of learning. Since the generator initially produces mostly incorrect solutions which leads to most $r_{g,\text{out}}(\mathbf{o}_g^i) = 0$, the majority of verifier supervision is biased toward negative labels. If we directly apply the original GRPO advantage calculation (as used for the generator in Eq. (6)), we find that the verifier quickly collapses to always predicting $y_{\text{final}} = 0$, resulting in a trivial but locally stable solution. This collapse not only harms verifier performance but also provides poor step-level verification reward signals to the generator, degrading overall co-training dynamics. To mitigate this issue, we introduce a class-aware reweighting scheme into the verifier's advantage computation. Specifically, we apply a sample-specific scaling factor $s_+$ or $s_-$ based on the correctness of the corresponding generator solution after normalizing the outcome rewards using the group $\mathbf{R}_{v,\text{out}} = \{r_{v,\text{out}}(\mathbf{o}_v^i)\}_{i=1}^M$ statistics:

$$\hat{A}_{v,t}^i = s_i \times \frac{r_{v,\text{out}}(\mathbf{o}_v^i) - \text{mean}(\mathbf{R}_{v,\text{out}})}{\text{std}(\mathbf{R}_{v,\text{out}})}, \quad s_i = \begin{cases} s_+ \in \mathbb{R}^+, & \text{if } r_{g,\text{out}}(\mathbf{o}_g^i) = 1, \\ s_- \in \mathbb{R}^+, & \text{otherwise.} \end{cases} \quad (9)$$

The coefficients $\{s_+, s_-\}$ are set to be inversely proportional to the square root of the number of samples with correct and incorrect generator outputs, respectively. In practice, we maintain these values per batch using an exponential moving average (EMA) to ensure smooth updates throughout training. To build intuition for Eq. (9), consider the case where most rewards $r_{g,\text{out}}(\mathbf{o}_g^i) = 0$, which results in $s_+ > s_-$. Under this condition, Eq. (9) effectively amplifies the contribution of the relatively fewer samples with $r_{g,\text{out}}(\mathbf{o}_g^i) = 1$ in the overall objective Eq. (1), while downweighting the influence of the more frequent samples with $r_{g,\text{out}}(\mathbf{o}_g^i) = 0$.

**Remarks.** We highlight three key advantages of TANGO. First, our verifier is trained via RL, enjoying stronger reasoning and generalization capabilities without requiring costly step-level annotations. Second, unlike prior logit-based methods, it produces transparent, text-based judgments that reduce step-level noise and introduce sampling stochasticity, mitigating reward hacking. Third, the evolving generator produces increasingly diverse outputs, exposing the verifier to broader reasoning patterns and encouraging it to adapt new verification strategies, which in turn improves the generator.

## 4 Experiments

**Base models.** We primarily evaluate our method on mathematical tasks to assess reasoning capability, and on unseen out-of-domain tasks to measure generalization. The generator uses Qwen2.5-Math-7B [59] for its strong mathematical reasoning, while the verifier uses Qwen2.5-7B [58] due to its larger context window accommodating both questions and generator outputs. Notably, Qwen2.5-7B underperforms on math tasks, making our verifier initially weaker than the generator, unlike prior work relying on stronger verifiers for distillation. Instead, our framework uses mutual reinforcement, enabling both agents to co-evolve from weaker starts, yielding a more scalable and practical solution.

**Implementation details.** We first perform SFT on the generator using 113K math prompts from Eurus-2-SFT-Data [12], guiding step-by-step reasoning enclosed in step tags. Responses are produced using Llama-3.1-70B-Instruct [16] with a system prompt (see Appendix F). The verifier is initialized directly from the base model without SFT. In the RL stage, we use 455K math question–answer pairs from Eurus-2-RL-Data [12]. We set $N_g = 3$ and $N_v = 1$, i.e., the verifier updates once every three generator steps, to compensate for slower generation optimization. The generator is trained for 200 steps by default (300 for Table 2 comparison). To prevent early instability, we warm up the verifier for 40 steps to learn the output formatting and reach a reasonable accuracy before guiding the generator.

Experiments are conducted using veRL [45], with 5 rollouts per prompt. Both generator and verifier policies are trained using AdamW [34] with a constant learning rate $1 \times 10^{-6}$, batch size 256, and microbatch size 4. $\gamma$ is set to 0.8. The coefficient $\alpha$ follows an exponential decay schedule,

Table 1: **Performance comparison of TANGO with different vanilla RL algorithms** on mathematical and out-of-domain reasoning benchmarks. TANGO consistently yields substantial improvements across all tasks when combined with various RL algorithms. All RL models are trained for 200 generator steps.

| Model | Mathematical Reasoning | | | | | | Out-of-Domain Reasoning | | | | |
| | MATH500 | AIME2024 | AIME2025 | AMC2023 | OlympiadBench | Avg. | BGQA | CRUXEval | StrategyQA | TableBench | Avg. |
|---|---|---|---|---|---|---|---|---|---|---|---|
| TANGO-7B-SFT | 66.6 | 3.3 | 3.3 | 27.5 | 28.1 | 25.8 | 46.6 | 44.3 | 85.9 | 34.4 | 52.8 |
| GRPO | 74.6 | 13.3 | 10.0 | 50.0 | 36.9 | 37.0 | 55.3 | 48.8 | 88.1 | 38.2 | 57.6 |
| **GRPO w/ TANGO** | **81.4** | **20.0** | **20.0** | **65.0** | **43.9** | **46.1** | **60.5** | **51.4** | **90.0** | **42.3** | **61.1** |
| RLOO | 74.0 | 13.3 | 10.0 | 52.5 | 36.0 | 37.2 | 55.0 | 48.5 | 87.5 | 38.6 | 57.4 |
| **RLOO w/ TANGO** | **80.8** | **23.3** | **16.7** | **67.5** | **45.3** | **46.7** | **60.9** | **52.9** | **90.4** | **43.0** | **61.8** |
| REINFORCE++ | 73.2 | 13.3 | 10.0 | 52.5 | 36.7 | 37.1 | 53.2 | 47.8 | 87.3 | 39.8 | 57.0 |
| **REINFORCE++ w/ TANGO** | **81.6** | **20.0** | **23.3** | **65.0** | **44.6** | **46.9** | **61.1** | **52.0** | **89.0** | **44.2** | **61.6** |

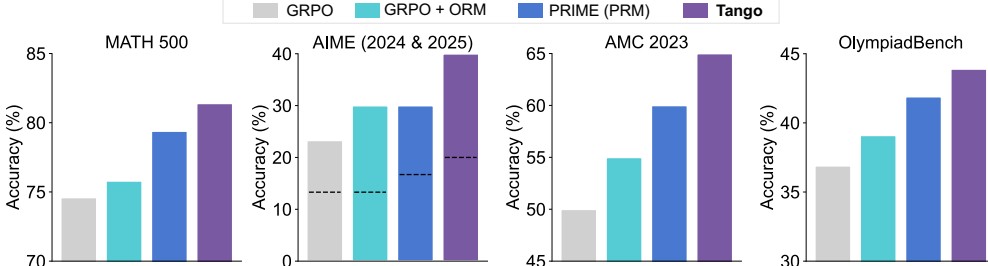

Figure 3: **Performance comparison of TANGO with ORM- and PRM-based baselines.** TANGO consistently outperforms models guided by ORM or PRM, demonstrating the superiority of our co-evolving framework in boosting generator's performance. For AIME, results for 2024 and 2025 are combined and shown below and above the dashed line respectively. All models are trained for 200 generator steps. We reproduce and evaluate PRIME at 200 steps, achieving better performance than the 240-step results reported in its original paper [12].

starting at 0.1 for GRPO and RLOO, and 0.5 for REINFORCE++, to balance step- and outcome-level advantages with an emphasis on step-level guidance early in training. These relatively high initial values of $\alpha$ encourage step rewards to guide early exploration more, while gradually decaying to $1 \times 10^{-3}$ to reduce reward hacking risks in later training. All baselines share identical generator SFT and RL configurations for fair comparison, with GRPO used as the default RL algorithm unless otherwise specified. Please refer to Appendix D for more implementation details.

**Benchmark and evaluation.** We primarily evaluate the generator on five competition-level math benchmarks: AIME 2025 [38], AIME 2024 [3], AMC 2023 [4], MATH-500 [32], and Olympiad-Bench [20]. Following [44], we further assess general reasoning and generalization capabilities using four out-of-domain benchmarks: logical reasoning (BoardgameQA, i.e., BGQA [26]), code reasoning (CRUXEval [17]), commonsense reasoning (StrategyQA [15]), and tabular reasoning (TableBench [56]). All models, including baselines, are evaluated via greedy decoding, reporting zero-shot pass@1 accuracy, i.e., the percentage of problems correctly solved on the first attempt. Additionally, we evaluate our verifier's step-level verification accuracy on ProcessBench [64], which contains annotated reasoning errors for competition-level math problems.

## 4.1 Main Results

**Comparison with vanilla RL post-training methods.** We first evaluate TANGO on standard RL algorithms commonly used for LLM post-training – GRPO [43], RLOO [1], and REINFORCE++ [21] – comparing each against its vanilla counterpart, which employs rule-based outcome rewards. The generator's performance after SFT is also included for reference. As shown in Table 1, integrating TANGO consistently yields substantial improvements across all benchmarks, particularly on challenging math competitions. For example, TANGO with GRPO achieves relative gains of 50.4% on AIME 2024, 100.0% on AIME 2025, and 30.0% on AMC 2023, averaging a 24.6% improvement across all math tasks. Furthermore, TANGO with GRPO enhances generalization to out-of-domain reasoning tasks, delivering an average relative improvement of 6.1%. Please refer to Appendix E for more results using the Llama base model.

Similar trends occur with RLOO and REINFORCE++, often surpassing those seen with GRPO: RLOO achieves relative gains averaging 25.5% on math and 7.7% on out-of-domain tasks, while

Table 2: **System-level performance comparison with prior methods** on mathematical and out-of-domain reasoning benchmarks. TANGO achieves state-of-the-art performance among 7B/8B-scale models across both domains. For math reasoning, results are from the original papers or prior work [18, 44], except PRIME [12], which we reproduce and evaluate, finding it outperforms the best 592-step results reported in the original paper, and for AIME 2025, which we evaluate for all methods. For out-of-domain reasoning, results are from [44]. Our TANGO-Qwen-7B is trained for 300 steps. Best performance per task among 7B/8B models is bolded.

| Model | Mathematical Reasoning | | | | | | Out-of-Domain Reasoning | | | | |
| --- | --- | --- | --- | --- | --- | --- | --- | --- | --- | --- | --- |
| | MATH500 | AIME2024 | AIME2025 | AMC2023 | OlympiadBench | Avg. | BGQA | CRUXEval | StrategyQA | TableBench | Avg. |
| *Frontier LLMs* | | | | | | | | | | | |
| GPT-4o [22] | 76.6 | 9.3 | - | 47.5 | 43.3 | - | - | - | - | - | - |
| Claude3.5-Sonnet [5] | 78.3 | 16.0 | - | - | - | - | - | - | - | - | - |
| o1-preview [23] | 85.5 | 44.6 | - | 90.0 | - | - | - | - | - | - | - |
| o1-mini [23] | 90.0 | 56.7 | - | 95.0 | 65.3 | - | - | - | - | - | - |
| *Open-sourced reasoning LLMs (large)* | | | | | | | | | | | |
| Llama-3.1-70B-Instruct [16] | 68.0 | 13.3 | - | 42.5 | 29.4 | - | 58.3 | 59.6 | 88.8 | 34.2 | - |
| OpenMath2-Llama3.1-70B [51] | 71.8 | 13.3 | - | 45.0 | 30.1 | - | 68.7 | 35.1 | 95.6 | 46.8 | - |
| NuminaMath-72B-CoT [6] | 64.0 | 3.3 | - | 70.0 | 32.6 | - | - | - | - | - | - |
| Qwen2.5-Math-72B-Instruct [59] | 82.6 | 23.3 | - | 70.0 | 49.0 | - | - | - | - | - | - |
| QwQ-32B-Preview [50] | 90.6 | 50.0 | 33.3 | 77.5 | 61.2 | 62.5 | 71.1 | 65.2 | 88.2 | 51.5 | 69.0 |
| *Open-sourced reasoning LLMs (small)* | | | | | | | | | | | |
| Llama-3.1-8B-Instruct [16] | 51.9 | 3.3 | 3.3 | 22.5 | 15.1 | 19.2 | 50.3 | 38.5 | **92.2** | 32.4 | 53.4 |
| OpenMath2-Llama3.1-8B [51] | 67.8 | 6.7 | 3.3 | 37.5 | 28.9 | 28.8 | 49.0 | 11.1 | 84.4 | 34.2 | 44.7 |
| Qwen2.5-7B-Instruct [58] | 75.5 | 10.0 | 6.7 | 52.5 | 35.5 | 36.0 | 53.0 | **58.1** | 91.3 | 43.2 | 61.4 |
| Qwen2.5-Math-7B-Instruct [59] | **83.6** | 16.7 | 10.0 | 62.5 | 41.6 | 42.9 | 51.3 | 28.0 | 85.3 | 36.2 | 50.2 |
| rStar-Math-7B [18] | 78.4 | **26.7** | - | 47.5 | **47.1** | - | - | - | - | - | - |
| Eurus-2-7B-PRIME [12] | 80.4 | **26.7** | 13.3 | 60.0 | 43.7 | 44.8 | - | - | - | - | - |
| *Ours* | | | | | | | | | | | |
| **TANGO-Qwen-7B** | 82.4 | **26.7** | **23.3** | **70.0** | 45.3 | **49.5** | **62.3** | 54.0 | 91.4 | **43.6** | **62.8** |

REINFORCE++ obtains gains of 26.4% and 8.1%, respectively. These results highlight TANGO's robustness and broad applicability across diverse RL algorithms and tasks.

We further visualize TANGO's training dynamics in Figure 1. Notably, our method matches the accuracy of vanilla GRPO after 200 generator steps in only 60 steps, a 3.3× improvement in training efficiency (the figure plots global steps to account for both the generator and verifier). At 200 generator steps, TANGO also achieves a 9.1% higher relative accuracy, underscoring significant gains in both training efficiency and reasoning quality.

**Comparison with different RM baselines.** We compare TANGO with ORM and PRM baselines in Figure 3. For PRM, we select PRIME [12] as it similarly does not require step-level supervision, making it directly comparable to our method. Note that our method uses the same SFT and RL data as PRIME, as well as the same base model. Integrated with GRPO, TANGO substantially outperforms both ORM and PRIME across all benchmarks. We attribute these gains to our co-evolving design, where the generator and verifier mutually reinforce each other through interleaved RL training. Unlike ORM, which provides only sparse, outcome-level feedback, our verifier delivers detailed, step-level rewards, guiding the generator toward better reasoning. Compared to PRIME, our RL-trained verifier offers more accurate and robust reasoning. Its generative, sampling-based verification introduces stochasticity and enables longer chains of thought, resulting in rewards that are more resistant to hacking and better aligned with true correctness, providing stronger and more informative supervision.

**System-level comparison with prior methods.** We further validate TANGO through a comprehensive system-level comparison against previous methods on mathematical and out-of-domain reasoning benchmarks, as shown in Table 2. Among 7B/8B-scale reasoning LLMs, TANGO achieves state-of-the-art performance, averaging 49.5% accuracy on math tasks and 62.8% on out-of-domain tasks. The improvements are especially significant on challenging math competitions, with scores of 26.7% on AIME 2024, 23.3% on AIME 2025, and 70.0% on AMC 2023, surpassing all prior models at similar scales. These gains highlight the effectiveness of our co-evolving training framework, where the generator and verifier mutually reinforce each other through progressive refinement of feedback, enabling deeper exploration and improved reasoning capabilities on complex problems.

## 4.2 Verifier Results of TANGO

In the previous section, we demonstrated that TANGO delivers a strong generator through co-evolving, interleaved RL training. Here, we show that the verifier also significantly benefits from this co-evolution, steadily improves throughout training and ultimately becomes highly effective.

Table 3: **Evaluation results on ProcessBench.** The verifier of TANGO achieves state-of-the-art performance among 7B/8B-scale models without using any process labels. The metric reported is the F1 score of the respective accuracies on erroneous and correct samples. Best performance per dataset among 7B/8B models is bolded.

| Model | GSM8K | MATH | OlympiadBench | Omni-MATH | Avg. |
|---|---|---|---|---|---|
| *Open-sourced language models, prompted as critic models* | | | | | |
| Qwen2.5-32B-Instruct [58] | 65.6 | 53.1 | 40.0 | 38.3 | 49.3 |
| Llama-3.1-70B-Instruct [16] | 74.9 | 48.2 | 46.7 | 41.0 | 52.7 |
| Qwen2.5-Math-72B-Instruct [59] | 65.8 | 52.1 | 32.5 | 31.7 | 45.5 |
| Llama-3.1-8B-Instruct [16] | 10.9 | 5.1 | 2.8 | 1.6 | 5.1 |
| Qwen2.5-Math-7B-Instruct [59] | 26.8 | 25.7 | 14.2 | 12.7 | 19.9 |
| Qwen2.5-7B-Instruct [58] | 36.5 | 36.6 | 29.7 | 27.4 | 32.6 |
| *Open-sourced process reward models (PRMs)* | | | | | |
| Math-Shepherd-PRM-7B [54] | 47.9 | 29.5 | 24.8 | 23.8 | 31.5 |
| RLHFlow-PRM-Mistral-8B [57] | 50.4 | 33.4 | 13.8 | 15.8 | 28.4 |
| RLHFlow-PRM-Deepseek-8B [57] | 38.8 | 33.8 | 16.9 | 16.9 | 26.6 |
| EurusPRM-7B [12] | 56.6 | 43.0 | 27.3 | 26.8 | 35.1 |
| Skywork-PRM-7B [36] | **70.8** | **53.6** | 22.9 | 21.0 | 42.1 |
| *Our verifier* | | | | | |
| **TANGO-Qwen-7B (verifier)** | 53.1 | 48.2 | **37.8** | **36.3** | **43.9** |

We first visualize the verifier's final verification F1 score over training steps in Figure 1, observing consistent improvement. Although the absence of gold step-level labels in our math training dataset prevents direct tracking of step-wise accuracy, we provide such analysis using a well-designed algorithmic reasoning task with step-level annotations in Section 4.3. There, we confirm that RL training enhances both step-level and final verification performance throughout the training.

We further evaluate the step-level verification accuracy of our final verifier on ProcessBench [64], a benchmark featuring competition-level math problems annotated with step-wise reasoning errors. As shown in Table 3, TANGO's verifier achieves state-of-the-art results among 7B/8B-scale models, despite training without any step-level supervision. It notably excels on the most challenging subsets, OlympiadBench and Omni-MATH, surpassing previous models significantly, even outperforming the much larger Qwen2.5-Math-72B-Instruct, despite being initiated only from a Qwen2.5-7B base.

These results confirm that our verifier progressively improves both its outcome-level (Figure 1) and step-level verification accuracy (Section 4.3) over the course of co-evolving RL training with the generator. Ultimately, it delivers highly accurate step-level verification even on the most challenging mathematical problems (Table 3).

## 4.3   Ablation Analysis with Gold Step-Level Information

In this section, we design an algorithmic reasoning task with gold step-level labels to enable a detailed analysis of TANGO and better illustrate the co-evolution dynamics between the generator and the verifier. Specifically, we adopt the last letter concatenation problem introduced in [55]. The prompt is constructed to elicit step-by-step reasoning from the generator, where the $n$-th step involves extracting the last letter of the $n$-th word (see Appendix F for examples). This setup allows us to automatically generate gold step-level outputs without any additional annotation effort when constructing the training and evaluation datasets, and also enables evaluation of the verifier's step-level judgments. We use Qwen2.5-1.5B [58] as the base model for both the generator and the verifier. We compare TANGO against three baselines: (i) the vanilla GRPO method without a verifier, (ii) GRPO with TANGO while keeping the generator fixed, and (iii) GRPO with TANGO while keeping the verifier fixed. More detailed experiment setups can be found in Appendix D.

**TANGO (ours).**   As shown in Figure 4, when both the generator and verifier are jointly updated under the TANGO framework, we observe consistent and strong improvements. The generator achieves the best accuracy (left), while the verifier steadily improves on both step-level and outcome-level F1 scores (middle and right). This result confirms that although the verifier is trained only with outcome-level rewards, it gradually improves its step-level verification accuracy as RL enhances its chain-of-thought reasoning. It also demonstrates that the generator and verifier mutually reinforce each other, leading to stronger reasoning capabilities and more accurate verification.

**Fixing generator.**   In this setting, only the verifier is updated. Initially, it learns from the fixed generator's output distribution and improves its F1 score. However, since the generator remains static,

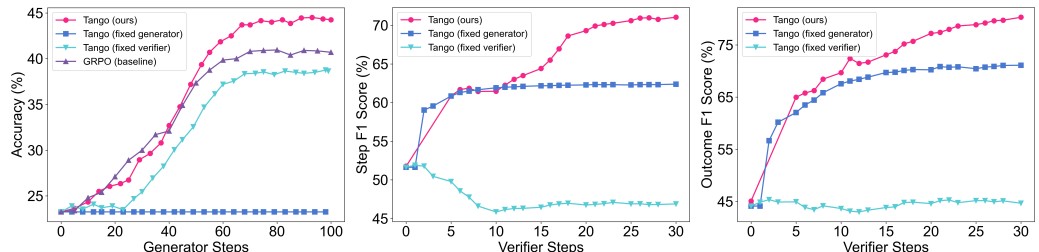

Figure 4: **Ablation of generator and verifier training dynamics in the algorithmic reasoning task.** Left: generator accuracy v.s. generator training steps; Middle: verifier step F1 score v.s. verifier training steps; Right: verifier outcome F1 score v.s. verifier training steps. All curves are evaluated on unseen test data.

the verifier's progress quickly plateaus, as shown in the middle and right panels. This underscores the importance of continuously improving the generator to provide richer and more diverse reasoning traces that can better support verifier training.

**Fixing verifier.** Although the verifier is frozen, its F1 scores (middle and right) shift slightly as generator training alters its input distribution during evaluation. On the generator side (left), performance remains flat for the first 20 steps due to inaccurate step-level feedback from the fixed verifier, which misguides learning. As the $\alpha$ schedule gradually shifts focus from misleading step-level to reliable gold outcome-level rewards, the generator starts to improve. However, its final accuracy still lags behind the baseline, highlighting how static and inaccurate verifier feedback can hinder learning, especially early on, when step-level signals are most critical for strategy exploration.

# 5 Conclusions

We present TANGO, a novel unified RL-based framework that jointly trains an LLM generator and a generative, process-level verifier using RL in an interleaved manner. Unlike existing approaches that rely on frozen or SFT-trained reward models, TANGO is the first to train the verifier via RL and co-evolve it with the generator without requiring any process-level annotations. Extensive experiments show that both the generator and verifier of TANGO, through mutual reinforcement, achieve state-of-the-art performance across multiple challenging reasoning benchmarks.

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

# A  Broader Impacts

We introduce a novel framework that jointly trains a generator and a verifier via interleaved RL, with the goal of enhancing the reasoning capabilities of language models through mutual reinforcement. While we believe this approach can meaningfully improve the reliability, interpretability, and generalization of LLM reasoning, several potential societal risks associated with such advancements warrant careful consideration.

First, as models become increasingly proficient at producing fluent and logically structured multi-step outputs, there is a risk that such capabilities may be applied in settings where correctness is difficult to ascertain, or where persuasive yet flawed reasoning can lead to harmful outcomes, for instance, in domains like education, policymaking, or scientific communication. Second, although the verifier provides step-level natural language feedback that is more transparent and less noisy than scalar reward signals, such responses may still be perceived as authoritative even when incorrect, particularly by non-expert users. Third, the use of RL in open-ended generation remains imperfectly understood, and may introduce unintended behaviors through reward misalignment or policy over-optimization.

That said, we note that these concerns are not unique to our approach, but reflect broader challenges in RL for LLM reasoning. Our design choices, such as employing generative, text-based verification and sampling-based reward signals, may help mitigate some known issues related to rigidity and reward hacking. Moreover, by training the verifier through RL rather than supervised imitation, our framework encourages the verifier to develop more accurate and adaptive verification strategies, which may help reduce the likelihood of producing misleading or unjustified assessments. Nonetheless, we encourage responsible and cautious use of this framework, particularly in domains where reasoning quality directly impacts users or high-stakes decisions.

# B  Limitations

While TANGO demonstrates strong performance across a range of reasoning tasks, several aspects merit further discussion. First, the co-evolving RL training in TANGO offers a higher performance ceiling compared to single-agent RL approaches, but realizing this potential requires careful tuning, such as generator–verifier update schedule and advantage mixing strategy. While the framework remains effective with minimal tuning, achieving optimal performance relies on well-coordinated training dynamics. Second, jointly training the generator and verifier in TANGO introduces additional computation compared to optimizing the generator alone, but the overhead is modest in practice. Much like verifying a mathematical proof is easier than constructing one, the verifier's task is more sample-efficient and easier to optimize than generation, allowing less frequent updates. We empirically find that updating the verifier once every three generator steps strikes a good balance between performance and efficiency, keeping the overall cost comparable to single-agent setups. Lastly, while our evaluation spans both mathematical and several structured out-of-domain reasoning tasks, including logic, code, commonsense, and tabular reasoning, it does not yet cover more open-ended domains such as long-context reasoning, multi-hop fact verification, or dialogue-based reasoning. We expect TANGO can generalize to such settings, which we reserve for future work.

# C TANGO Algorithm Flow

The algorithm flow of TANGO is detailed in Algorithm 1.

---

**Algorithm 1:** Interleaved RL Training of Generator and Verifier in TANGO

---

**Input:** Training data distribution $\mathcal{P}$, generator policy $\pi_g$, verifier policy $\pi_v$, mixing weight $\alpha$,
   rollout size $M$, generator update steps $N_g$, verifier update steps $N_v$
**Output:** Trained generator $\pi_g$ and verifier $\pi_v$
**while** *not converged* **do**

    **for** $n = 1$ **to** $N_g$ **do**

        Sample a batch $\mathcal{B} \sim \mathcal{P}$. For each $(\mathbf{x}, \mathbf{y}) \in \mathcal{B}$, generate $M$ rollouts of multi-step solutions
        $\{\mathbf{o}_g^i\}_{i=1}^M \sim \pi_g(\cdot \mid \mathbf{x})$ and query the verifier to generate corresponding verifications:
        $\mathbf{o}_v^i \sim \pi_v(\cdot \mid \mathbf{x}, \mathbf{o}_g^i)$, $i = 1, 2, \ldots, M$.
        Extract predicted answer $\hat{\mathbf{y}}^i$ from $\mathbf{o}_g^i$ and step-level judgments $\{y_{\text{step},k}^i\}_{k=1}^{K^i}$ from $\mathbf{o}_v^i$.
        Compute generator advantages $\{\hat{A}_{g,t}^i\}_{i=1}^M$ via Eq. (7). Perform policy gradient update
        on generator $\pi_g$ using $\hat{A}_{g,t}^i$.

    **end**

    **for** $n = 1$ **to** $N_v$ **do**

        Sample a batch $\mathcal{B} \sim \mathcal{P}$. For each $(\mathbf{x}, \mathbf{y}) \in \mathcal{B}$, generate a multi-step solution
        $\mathbf{o}_g \sim \pi_g(\cdot \mid \mathbf{x})$ and query the verifier to generate $M$ verification rollouts:
        $\{\mathbf{o}_v^i\}_{i=1}^M \sim \pi_v(\cdot \mid \mathbf{x}, \mathbf{o}_g)$, $i = 1, 2, \ldots, M$.
        Extract final judgment $y_{\text{final}}^i$ from $\mathbf{o}_v^i$. Compute verifier advantages $\{\hat{A}_{v,t}^i\}_{i=1}^M$ via Eq. (9).
        Perform policy gradient update on verifier $\pi_v$ using $\hat{A}_{v,t}^i$.
        Update EMA reweighting coefficients $s_+$ and $s_-$.

    **end**

**end**

---

# D Additional Experiment Details

## D.1 Main Experiments

In addition to the experimental setup described in Section 4, we provide further details below.

For the SFT stage, we first generate training data by prompting Llama-3.1-70B-Instruct (system prompt shown in Section F.1) with a decoding temperature of 0.1 and a top-$p$ value of 0.5. The generated responses are then used to perform SFT on the generator base model Qwen2.5-Math-7B. We conduct full-parameter SFT using a learning rate of $5 \times 10^{-6}$ with the AdamW optimizer, a cosine annealing learning rate schedule, and a warmup ratio of 0.1. The model is trained for 800 steps with a batch size of 64.

For the RL stage, both the generator and verifier generate rollouts using a sampling temperature of 1.0 and a top-$p$ value of 1.0. We set the KL loss coefficient $\beta$ to 0.001. The EMA decay factor for tracking correct and incorrect samples from the generator is set to 0.8.

## D.2 Algorithmic Reasoning Experiment

For the algorithmic reasoning task, specifically, the last-letter concatenation experiment presented in Section 4.3, we first construct SFT datasets by randomly generating 2 to 4 words, each containing 3 to 6 characters. These datasets are used to train both the generator and verifier for several dozen steps, primarily to ensure that the Qwen-2.5-1.5B base models learns to follow the specified instructions and produce outputs that conform to the required format. For RL training dataset, we similarly generate input sequences consisting of 2 to 10 words, with each word containing 3 to 10 characters. The test dataset is constructed in the same manner but includes slightly longer sequences, 2 to 12 words with 3 to 12 characters, to cover the evaluation of the model's out-of-distribution generalization ability. Most training hyperparameters follow those used in the main experiments detailed in Section 4 and Appendix D.1, except that we use a batch size of 64 and an exponential learning rate decay schedule for the generator. For the three baseline settings, vanilla GRPO, fixing the generator, and fixing the verifier, we adopt the same configurations to ensure a fair comparison.

# E    Additional Results Using the Llama Base Model

To further demonstrate the generalizability of TANGO, we include results from a Llama base model in addition to the Qwen series of models. Specifically, we conducted experiments using Llama-3.1-8B-Instruct [16] as the base model for both the generator and verifier, following the same training and evaluation protocol as in Table 1. The results are presented in the table below.

Table 4: **TANGO performance using Llama-3.1-8B-Instruct as the base model** on mathematical benchmarks. TANGO still achieves substantial improvements across datasets on Llama base models.

| Model | MATH500 | AIME2024 | AIME2025 | AMC2023 | OlympiadBench | Avg. |
|---|---|---|---|---|---|---|
| Llama3.1-8B-Instruct (Base) | 46.4 | 3.3 | 3.3 | 22.5 | 13.6 | 17.9 |
| TANGO-Llama3.1-8B-SFT | 49.4 | 3.3 | 3.3 | 25.0 | 15.0 | 19.2 |
| GRPO | 56.2 | 10.0 | 3.3 | 35.0 | 20.9 | 25.1 |
| **GRPO w/ TANGO** | **60.5** | **13.3** | **6.7** | **40.0** | **23.6** | **28.8** |

These results show that even when using Llama as the base model, TANGO continues to deliver significant improvements, demonstrating strong generalization across different model families. This further confirms that TANGO's effectiveness stems from our interleaved RL co-evolving framework, rather than being specific to any particular model family.

# F    Additional Generation and Verification Examples

## F.1    Mathematical Reasoning

> **Generator's Prompt in Math Reasoning**
>
> ```
> You are a helpful Assistant that solves mathematical problems step-by-step.
> Your task is to provide a detailed solution process within specific tags.
>
> You MUST follow this exact format:
> 1. Start with a <think> section containing your step-by-step reasoning.
> 2. Inside <think>, each distinct logical step MUST be enclosed in its own <step> </step>
> tags.
> 3. After <think>, provide the final answer within <answer> </answer> tags, using the \boxed
> {} format.
>
> Here is an example of the required format:
>
> User: Calculate 15 - (3 * 2).
> Assistant: <think>
> <step>First, calculate the expression inside the parentheses, which is 3 multiplied by 2.</
> step>
> <step>3 * 2 equals 6.</step>
> <step>Next, subtract the result from the original number, which is 15 minus 6.</step>
> <step>15 - 6 equals 9.</step>
> </think>
> <answer>\boxed{9}</answer>
>
> You MUST strictly adhere to this format.
> - Output ONLY the content within <think>, <step>, and <answer> tags.
> - Do NOT include any text or characters before the opening <think> tag or after the closing
>  </answer> tag.
> - Ensure every part of your step-by-step reasoning is inside <step> </step> tags within the
>  <think> </think> section.
> - Provide the final answer inside <answer>\boxed{}</answer>. Your final answer will be
> extracted automatically by the \boxed{} tag.
> ```
>
> User: **For how many integers $n$ does the expression**
>
> $$\sqrt{\frac{\log(n^2) - (\log n)^2}{\log n - 3}}$$
>
> **represent a real number, where log denotes the base $10$ logarithm?**
> Assistant:

## Generator's Response in Math Reasoning

<think>
<step>First, simplify the expression inside the square root: $\log(n^2) - (\log n)^2$.</step>
<step>Using the logarithm property $\log(n^2) = 2\log n$, the expression becomes $2\log n - (\log n)^2$.</step>
<step>So the expression inside the square root is $\frac{2\log n - (\log n)^2}{\log n - 3}$.</step>
<step>Factor the numerator: $2\log n - (\log n)^2 = \log n(2 - \log n)$.</step>
<step>The expression now is $\sqrt{\frac{\log n(2 - \log n)}{\log n - 3}}$.</step>
<step>For the expression to be a real number, the fraction $\frac{\log n(2 - \log n)}{\log n - 3}$ must be non-negative and $\log n \neq 3$.</step>
<step>Consider the critical points where the numerator or the denominator is zero: $\log n = 0$, $\log n = 2$, and $\log n = 3$.</step>
<step>$\log n = 0$ implies $n = 1$.</step>
<step>$\log n = 2$ implies $n = 100$.</step>
<step>$\log n = 3$ implies $n = 1000$.</step>
<step>Now, analyze the intervals determined by these critical points: $(0, 1)$, $(1, 2)$, $(2, 3)$, $(3, \infty)$.</step>
<step>For $(0, 1)$: $0 < \log n < 1$ implies $\log n(2 - \log n) > 0$, and $\log n - 3 < 0$, so the fraction is positive.</step>
<step>For $(1, 2)$: $0 < \log n < 2$ and $\log n - 3 < -1$, so the numerator and denominator must be either both positive or both negative. Since $0 < \log n - 3 < -1$, the fraction is always negative.</step>
<step>For $(2, 3)$: $1 < \log n < 2$ implies $\log n(2 - \log n) > 0$, and $\log n - 3 < -1$, so the fraction is negative.</step>
<step>For $(3, \infty)$: $\log n > 0$ and $\log n - 3 > 1$, so the numerator and denominator are both negative, making the fraction positive.</step>
<step>Thus, the expression is defined for $0 < n < 1$ and $100 < n < 1000$.</step>
<step>The integers in these intervals are $n = 1$ and $n = 101, 102, \ldots, 999$, which gives us $1 + 999$ integers.</step>
</think>
<answer> **1000** </answer>

You are a verification assistant specialized in mathematical reasoning. Your task is to carefully evaluate the provided solution step by step, checking for mathematical correctness and logical coherence. You will be given the original problem and the Assistant's solution, which contains a specific number of steps within <step> tags. You MUST verify EACH <step> block found in the Assistant's solution and provide your judgment using the exact format specified in the instructions. You MUST output ONLY the content within the specified verification tags and nothing else.

Here is the problem you need to verify, and the Assistant's solution:

**Problem:** For how many integers $n$ does the expression

$$\sqrt{\frac{\log(n^2) - (\log n)^2}{\log n - 3}}$$

represent a real number, where log denotes the base $10$ logarithm?

**Assistant's Solution:**
<think>
<step>First, simplify the expression inside the square root: $\log(n^2) - (\log n)^2$.</step>
<step>Using the logarithm property $\log(n^2) = 2\log n$, the expression becomes $2\log n - (\log n)^2$.</step>
<step>So the expression inside the square root is $\frac{2\log n - (\log n)^2}{\log n - 3}$.</step>
<step>Factor the numerator: $2\log n - (\log n)^2 = \log n(2 - \log n)$.</step>
<step>The expression now is $\sqrt{\frac{\log n(2 - \log n)}{\log n - 3}}$.</step>
<step>For the expression to be a real number, the fraction $\frac{\log n(2 - \log n)}{\log n - 3}$ must be non-negative and $\log n \neq 3$.</step>
<step>Consider the critical points where the numerator or the denominator is zero: $\log n = 0$, $\log n = 2$, and $\log n = 3$.</step>
<step>$\log n = 0$ implies $n = 1$.</step>
<step>$\log n = 2$ implies $n = 100$.</step>
<step>$\log n = 3$ implies $n = 1000$.</step>
<step>Now, analyze the intervals determined by these critical points: $(0,1)$, $(1,2)$, $(2,3)$, $(3,\infty)$.</step>
<step>For $(0,1)$: $0 < \log n < 1$ implies $\log n(2 - \log n) > 0$, and $\log n - 3 < 0$, so the fraction is positive.</step>
<step>For $(1,2)$: $0 < \log n < 2$ and $\log n - 3 < -1$, so the numerator and denominator must be either both positive or both negative. Since $0 < \log n - 3 < -1$, the fraction is always negative.</step>
<step>For $(2,3)$: $1 < \log n < 2$ implies $\log n(2 - \log n) > 0$, and $\log n - 3 < -1$, so the fraction is negative.</step>
<step>For $(3,\infty)$: $\log n > 0$ and $\log n - 3 > 1$, so the numerator and denominator are both negative, making the fraction positive.</step>
<step>Thus, the expression is defined for $0 < n < 1$ and $100 < n < 1000$.</step>
<step>The integers in these intervals are $n = 1$ and $n = 101, 102, \ldots, 999$, which gives us $1 + 999$ integers.</step>
</think>
<answer> $\boxed{1000}$ </answer>

The Assistant's solution contains 17 steps within <step> tags.

Please verify this solution step by step. For each of the 17 <step> blocks in the Assistant's Solution, you MUST provide ONE corresponding verification analysis within a <step> tag inside the <step_verification> section. After verifying all steps, provide a final overall judgment in the <final_verification> tag.

You MUST follow this exact format:

<step_verification>
<step>Step 1 Analysis: Your detailed verification reasoning goes here. Conclude with only one judgement: \boxed{CORRECT} or \boxed{INCORRECT}</step>
<step>Step 2 Analysis: Your detailed verification reasoning goes here. Conclude with only one judgement: \boxed{CORRECT} or \boxed{INCORRECT}</step>
... [CONTINUE for ALL 17 <step> blocks in the Assistant's Solution] ...
</step_verification>

<final_verification>\boxed{CORRECT} or \boxed{INCORRECT}</final_verification>

Here is an example:

Problem: What is 5 * 3 + 1?
Assistant's Solution:
<think>
<step>First, multiply 5 by 3. 5 * 3 = 15.</step>

```
<step>Then, add 1 to the result. 15 + 1 = 16.</step>
</think>
<answer>\boxed{16}</answer>

Your Verification:
<step_verification>
<step>Step 1 Analysis: The multiplication 5 * 3 is correctly calculated as 15. This step is
 mathematically sound. \boxed{CORRECT}</step>
<step>Step 2 Analysis: Adding 1 to the previous result (15) gives 16, which is correct.
This step follows logically and is mathematically accurate. \boxed{CORRECT}</step>
</step_verification>
<final_verification>\boxed{CORRECT}</final_verification>

IMPORTANT INSTRUCTIONS (Read Carefully):
1. The Assistant's solution has 17 steps. You MUST analyze and provide a verification for
EACH and EVERY one of these steps. The number of <step> tags within your <step_verification>
 section MUST be exactly 17.
2. You MUST analyze the step and provide YOUR OWN verification reasoning – DO NOT copy the
original solution text.
3. Each verification <step> must end with EXACTLY ONE judgement: either \boxed{CORRECT} or \
boxed{INCORRECT}.
4. Your final verification within <final_verification> must judge whether the overall
solution and final answer are correct.
5. You MUST output ONLY the content within the <step_verification> and <final_verification>
 tags. Do NOT output anything else.

Your Verification:
```

## Verifier's Response in Math Reasoning

<step_verification>
<step>Step 1 Analysis: The simplification of the expression inside the square root is necessary before proceeding further. The step is correctly identifying the need to simplify the expression. **CORRECT** </step>
<step>Step 2 Analysis: The logarithmic property $\log(n^2) = 2\log n$ is applied correctly, transforming the expression into $2\log n - (\log n)^2$. This step is mathematically sound. **CORRECT** </step>
<step>Step 3 Analysis: The expression inside the square root is correctly rewritten after applying the logarithmic property from Step 2. **CORRECT** </step>
<step>Step 4 Analysis: Factoring out $\log n$ from the numerator results in $\log n(2 - \log n)$, which is a valid algebraic operation. **CORRECT** </step>
<step>Step 5 Analysis: The expression is correctly simplified to $\sqrt{\frac{\log n(2-\log n)}{\log n-3}}$. This step follows logically from the previous steps. **CORRECT** </step>
<step>Step 6 Analysis: For the square root to represent a real number, the argument must be non-negative. Additionally, the denominator cannot be zero, meaning $\log n \neq 3$. This condition is correctly identified. **CORRECT** </step>
<step>Step 7 Analysis: Identifying the critical points where the numerator or denominator equals zero is crucial for determining the sign of the fraction. The critical points are correctly identified as $\log n = 0$, $\log n = 2$, and $\log n = 3$. **CORRECT** </step>
<step>Step 8 Analysis: When $\log n = 0$, it implies $n = 10^0 = 1$. This step is correct. **CORRECT** </step>
<step>Step 9 Analysis: When $\log n = 2$, it implies $n = 10^2 = 100$. This step is also correct. **CORRECT** </step>
<step>Step 10 Analysis: When $\log n = 3$, it implies $n = 10^3 = 1000$. This step is correct. **CORRECT** </step>
<step>Step 11 Analysis: The intervals determined by the critical points are correctly identified as $(0,1)$, $(1,2)$, $(2,3)$, and $(3,\infty)$. However, since $n$ must be an integer, the interval $(0,1)$ should not be considered. **INCORRECT** </step>
<step>Step 12 Analysis: For the interval $(0,1)$, the analysis is incorrect because $n$ cannot be less than 1 if it is an integer. The correct interval to consider would be $[1,1)$, which is empty. **INCORRECT** </step>
<step>Step 13 Analysis: For the interval $(1,2)$, the analysis is incorrect. It should state that the fraction is negative because the numerator is positive and the denominator is negative. **INCORRECT** </step>
<step>Step 14 Analysis: For the interval $(2,3)$, the analysis is incorrect. It should state that the fraction is positive because the numerator is negative and the denominator is negative. **INCORRECT** </step>
<step>Step 15 Analysis: For the interval $(3,\infty)$, the analysis is incorrect. It should state that the fraction is negative because the numerator is negative and the denominator is positive. **INCORRECT** </step>
<step>Step 16 Analysis: Based on the analysis of the intervals, the expression is defined for $100 < n < 1000$. The conclusion about the intervals is partially incorrect due to the misinterpretation of the first interval. **INCORRECT** </step>
<step>Step 17 Analysis: The count of integers in the intervals is incorrect based on the corrected intervals. There are no integers in $(0,1)$, and the correct count for $101 \leq n \leq 999$ is $999 - 101 + 1 = 899$. Including $n = 1$ incorrectly adds to the confusion. **INCORRECT** </step>
</step_verification>

<final_verification> **INCORRECT** </final_verification>

## F.2 Algorithmic Reasoning

---

**Generator's Prompt in Last Letter Concatenation**

```
A conversation between User and Assistant. The User asks a question, and the Assistant
solves it step by step.

The Assistant thinks through the problem in clear, logical steps, with each step enclosed
in <step> </step> tags within the overall <think> </think> section. Each <step> </step> tag
 pair should enclose ALL content for that step. After completing the thinking process, the
Assistant provides the final answer within <answer> </answer> tags.

The Assistant should strictly follow the format below:

<think>
<step>Your detailed reasoning. |Status|: Letters </step>
<step>Your detailed reasoning. |Status|: Letters </step>
...
</think>
<answer>\boxed{Your final letters}</answer>

User: You are given several words. Your task is to extract the last letter of each word and
 concatenate them to form a new word.

This is the problem: prepurpose, overflown, lovesick, burghalpenny, toran, outworn,
coadjacence, forgotten, imperilment, epidictical

Let us think step by step and please follow the format: <step> Your detailed reasoning. |
Status|: Letters </step>. In the n-th step, extract the last letters of the first n words
and replace Letters with the concatenation of these letters. Finally, return your result
using the following format: <answer>\boxed{your final letters}</answer>

You must output exactly 10 steps for this problem. Do not skip or add any extra steps.

Assistant:
```

---

**Generator's Response in Last Letter Concatenation**

```
<think>
<step>Extract the last letter from prepurpose, get e. |Status|: e</step>
<step>Extract the last letter from overflown, get d. |Status|: ed</step>
<step>Extract the last letter from lovesick, get k. |Status|: edk</step>
<step>Extract the last letter from burghalpenny, get y. |Status|: edky</step>
<step>Extract the last letter from toran, get n. |Status|: edkyn</step>
<step>Extract the last letter from outworn, get n. |Status|: edkynn</step>
<step>Extract the last letter from coadjacence, get e. |Status|: edkynne</step>
<step>Extract the last letter from forgotten, get d. |Status|: edkynned</step>
<step>Extract the last letter from imperilment, get m. |Status|: edkynnedm</step>
<step>Extract the last letter from epidictical, get l. |Status|: edkynnedml</step>
</think>
<answer> edkynnedml </answer>
```

## Verifier's Prompt in Last Letter Concatenation

You are a verification assistant specializing in algorithmic reasoning. Your task is to carefully evaluate each step in the provided solution for a last letter concatenation problem --- The problem involves concatenating the last letters of a list of words to form a new string. You must verify EACH numbered step in the solution and provide your judgment using the exact format specified in the instructions.

Here is the problem you need to verify, and the Assistant's solution:

**Problem:** prepurpose, overflown, lovesick, burghalpenny, toran, outworn, coadjacence, forgotten, imperilment, epidictical

**Assistant's Solution:**
```
<think>
<step>Extract the last letter from prepurpose, get e. |Status|: e</step>
<step>Extract the last letter from overflown, get d. |Status|: ed</step>
<step>Extract the last letter from lovesick, get k. |Status|: edk</step>
<step>Extract the last letter from burghalpenny, get y. |Status|: edky</step>
<step>Extract the last letter from toran, get n. |Status|: edkyn</step>
<step>Extract the last letter from outworn, get n. |Status|: edkynn</step>
<step>Extract the last letter from coadjacence, get e. |Status|: edkynne</step>
<step>Extract the last letter from forgotten, get d. |Status|: edkynned</step>
<step>Extract the last letter from imperilment, get m. |Status|: edkynnedm</step>
<step>Extract the last letter from epidictical, get l. |Status|: edkynnedml</step>
</think>
```

<answer> edkynnedml </answer>

The Assistant's solution contains 10 steps within <step> tags.

Please verify this solution step by step. For each of the 10 <step> blocks in the Assistant's Solution, you MUST provide ONE corresponding verification analysis within a <step> tag inside the <step_verification> section. After verifying all steps, provide a final overall judgment in the <final_verification> tag.

You MUST follow this exact format:

```
<step_verification>
<step>Step 1 Analysis: Your detailed verification reasoning goes here. Conclude with only one judgement: \boxed{CORRECT} or \boxed{INCORRECT}</step>
<step>Step 2 Analysis: Your detailed verification reasoning goes here. Conclude with only one judgement: \boxed{CORRECT} or \boxed{INCORRECT}</step>
... [CONTINUE for ALL 10 <step> blocks in the Assistant's Solution] ...
</step_verification>
<final_verification>\boxed{CORRECT} or \boxed{INCORRECT}</final_verification>
```

IMPORTANT INSTRUCTIONS (Read Carefully):
1. The Assistant's solution has 10 steps. You MUST analyze and provide a verification for EACH and EVERY one of these steps. The number of <step> tags within your <step_verification> section MUST be exactly 10.
2. You MUST analyze the step and provide YOUR OWN verification reasoning -- DO NOT copy the original solution text.
3. Each verification <step> must end with EXACTLY ONE judgement: either \boxed{CORRECT} or \boxed{INCORRECT}.
4. Your final verification within <final_verification> must judge whether the overall solution and final answer are correct.
5. You MUST output ONLY the content within the <step_verification> and <final_verification> tags. Do NOT output anything else.

Your Verification:

> **Verifier's Response in Last Letter Concatenation**
>
> ```
> <step_verification>
> <step>Step 1 Analysis: The last letter of prepurpose is e; Assistant gets e in its response.
> CORRECT </step>
> <step>Step 2 Analysis: The last letter of overflown is n; Assistant gets d in its response.
> INCORRECT </step>
> <step>Step 3 Analysis: The last letter of lovesick is k; Assistant gets k in its response.
> CORRECT </step>
> <step>Step 4 Analysis: The last letter of burghalpenny is y; Assistant gets y in its
> response. CORRECT </step>
> <step>Step 5 Analysis: The last letter of toran is n; Assistant gets n in its response.
> CORRECT </step>
> <step>Step 6 Analysis: The last letter of outworn is n; Assistant gets n in its response.
> CORRECT </step>
> <step>Step 7 Analysis: The last letter of coadjacence is e; Assistant gets e in its
> response. CORRECT </step>
> <step>Step 8 Analysis: The last letter of forgotten is n; Assistant gets d in its response.
> INCORRECT </step>
> <step>Step 9 Analysis: The last letter of imperilment is t; Assistant gets m in its
> response. INCORRECT </step>
> <step>Step 10 Analysis: The last letter of epidictical is l; Assistant gets l in its
> response. CORRECT </step>
> </step_verification>
> <final_verification> CORRECT </final_verification>
> ```

# G   Licenses for Assets

All models, datasets, and codebases used in our experiments are open-sourced and adopted under permissive research licenses, as detailed in Table 5.

Table 5: **Licenses for external assets** used in our experiments.

| Asset | Type | License / Terms |
|---|---|---|
| Qwen2.5-Math-7B [59] | Model | Apache 2.0 |
| Qwen2.5-7B [58] | Model | Apache 2.0 |
| Llama-3.1-70B-Instruct [16] | Model | Llama 3.1 Community License |
| Qwen2.5-1.5B [58] | Model | Apache 2.0 |
| Eurus-2-SFT-Data [12] | Dataset | MIT |
| Eurus-2-RL-Data [12] | Dataset | MIT |
| MATH 500 [32] | Dataset | MIT |
| AIME 2024 [3] | Dataset | Apache 2.0 |
| AIME 2025 [38] | Dataset | MIT |
| AMC 2023 [4] | Dataset | Apache 2.0 |
| OlympiadBench [20] | Dataset | Apache 2.0 |
| BoardgameQA [26] | Dataset | CC-BY 4.0 |
| CRUXEval [17] | Dataset | MIT |
| StrategyQA [15] | Dataset | MIT |
| TableBench [56] | Dataset | Apache 2.0 |
| ProcessBench [64] | Dataset | Apache 2.0 |
| veRL [45] | Codebase | Apache 2.0 |

