# OpenReview forum: "RL Tango: Reinforcing Generator and Verifier Together for Language Reasoning"
_NeurIPS.cc/2025/Conference — NeurIPS 2025 poster_

### Official Review · Reviewer_esdQ · 2025-07-02

**Clarity:** 3
**Significance:** 2
**Originality:** 3
**Rating:** 5
**Confidence:** 3

**Summary:**

Training language models with reinforcement learning on tasks with verifiable rewards (such as math and code) has emerged as a strong approach for improving reasoning in language models. Prior work typically relies on either a fixed verifier (e.g. rule based checks, trained reward model). The paper proposes an alternate paradigm where the verifier is a language models which is trained along with the generator. Specifically, the verifier and the generator are both trained with RL - the generator using feedback from the verifier and the verifier is trained using gold labels. Notably, even though the verifier provides step level feedback to the generator, it is trained only using the final label. Training the verifier and generator jointly can potentially mitigate reward hacking issues with fixed trained reward models and provide more fine-grained feedback than rule-based rewards. Empirical results show significant performance gains for the generator in the domain of math as well as generalization to some non-math domains. Additionally, even without process-level annotations the verifier demonstrate strong performance on ProcessBench. The authors also present some ablations to demonstrate the importance of joint training.

**Questions:**

* While reading the paper, I was reminded of two frameworks which I would be curious to hear the authors thoughts about:
    - The setup seems quite similar to the work on prover-verifier games (e.g. [1]) where you have a similar setup. This has also been instantiated in the language model setting [2]. To me the major difference seems the generative nature of the verifier training with RL which can provide process level feedback and using natural language reasoning.
    - Work on EM style training (e.g. [3]) where the "generative model" (i.e. M step) can be viewed as a verifier for the policy training (E-step). The difference from Tango is that the verifier in the EM case only provide the likelihood as the signal.
* Currently, the verifier and the generator both use different models. I am curious if you have tried setting them to be the same, or for example having both models be of different capabilities to understand what role that plays?
* For this setup, I believe an async training setup (e.g. [4,5]) could be quite useful to speed up training (even though authors say the computational overhead is not too high)
* The text explanation from the verifier is not used at all. Maybe it can be used for inference time search?


[1] Learning to Give Checkable Answers with Prover-Verifier Games. Anil et al. arXiv:2108.12099.

[2] Prover-Verifier Games improve legibility of LLM outputs. Kirchner et al. arXiv:2407.13692.

[3] Amortizing intractable inference in large language models. Hu et al. arXiv:2310.04363.

[4] Asynchronous RLHF: Faster and More Efficient Off-Policy RL for Language Models. Noukhovitch et al. arXiv:2410.18252.

[5] Trajectory Balance with Asynchrony: Decoupling Exploration and Learning for Fast, Scalable LLM Post-Training. Bartoldson et al. arXiv:2503.18929.

**Ethical Concerns:**

["NO or VERY MINOR ethics concerns only"]

**Final Justification:**

The authors added new experiments with different base models and addressed a lot of my concerns and comments. The paper is a strong contribution in the broad LLM + RL space.

**Limitations:**

Yes

**Quality:**

2

**Strengths And Weaknesses:**

Strengths:
* Generative verifiers have been shown to be quite helpful for improving RL training of LLMs. However, as noted in the paper reward hacking, fine-grained credit assignment and generalizability are challenges with static fixed verifiers. So overall I think the idea of jointly training the verifier and generator is quite interesting and promising. Joint training can conceptually mitigate some of these issues by updating the verifier to be in step with the generator.
* The paper also does a good job of clearly explaining the idea, and all of the details.
* To the best of my knowledge, the specific instantiation of the idea is also novel within the context of RL with language models (but I acknowledge that this is a fast moving area so I could be missing something).
* The method also seems to provide significant improvements empirically. The fact that the verifier can have such strong performance on ProcessBench without explicit process-level annotations is quite impressive. (Though as I discuss below the experiments can be improved)

Weaknesses:
* One major concern I have with the results presented in the paper is that they are limited to a Qwen2.5 base model. This model has some peculiarities [1] which can distort the evaluations. So it would be useful to have experiment on at least one model family (e.g. Gemma, Llama, OLMo, etc) to better understand the applicability of the method.
* (minor) While I think the results are impressive, I find the choice of models in Table 2 and Table 3 a bit arbitrary. The models all have different base models which can have a significant impact on the results, so I think these tables should be discussed with a bit more nuance.
* Finally, there no guarantees for the convergence of the joint training procedure. The authors mention that in their experiments it was not too sensitive. But I think a principle study could make the method even stronger.

Aside from the question of applicability with other base models I quite like the paper and lean towards accepting it.

[1] Spurious Rewards: Rethinking Training Signals in RLVR. Shao et al. arXiv:2506.10947.

---

> ### Author Rebuttal · Authors · 2025-07-31
>
> Dear Reviewer esdQ,
>
> We sincerely thank the reviewer for the thoughtful and detailed feedback. We are very delighted to see that the reviewer quite likes this paper and finds our idea novel and promising, our presentation clear, and our empirical performance strong. Below, we address the reviewer’s concerns in detail and provide additional clarifications where needed.
>
> > ***One major concern I have with the results presented in the paper is that they are limited to a Qwen2.5 base model … So it would be useful to have experiment on at least one model family...***
>
> We thank the reviewer for raising this important point. To better address the reviewer’s concern, we conducted additional experiments **using Llama-3.1-8B-Instruct as the base model**, following the same training/evaluation protocol as in Table 1. The results are presented in the table below.
>
> **Base Model: Llama-3.1-8B-Instruct**
> | Model            | MATH500 | AIME2024 | AIME2025 | AMC2023 | OlympiadBench | Avg. |
> |:-----------------|:-------:|:--------:|:--------:|:-------:|:--------------:|:----:|
> | TANGO-Llama3.1-8B-SFT     |  49.4   |   3.3    |   3.3    |  25.0   |     15.0       | 19.2 |
> | GRPO             |  56.2   |  10.0    |  3.3    |  35.0   |     20.9       | 25.1 |
> | **GRPO w/ TANGO**| **60.5**| **13.3** | **6.7** | **40.0**|   **23.6**     |**28.8**|
>
> They show that even when Llama is used as the base model, TANGO continues to deliver significant improvements, demonstrating strong generalization across different model families. This further confirms that TANGO’s effectiveness **stems from** our interleaved RL co-evolving framework, rather than being specific to any particular model architecture. We will include the above results and discussion in the revised paper.
>
> > ***I find the choice of models in Table 2 and Table 3 a bit arbitrary. The models all have different base models which can have a significant impact on the results, so I think these tables should be discussed with a bit more nuance.***
>
> Thank you for your suggestion. The purpose of Tables 2 and 3 is to position our results in the broader context of existing works and to demonstrate that TANGO-trained generators and verifiers remain highly competitive **regardless of base models**. For this reason, these tables include results from a variety of model families. However, we agree with the reviewer that more nuanced discussions would help readers better understand the performance of both our method and the baselines. In the revised paper, we will indicate the base model used for each method in Tables 2 and 3, and include additional discussions in the main text.
>
> Additionally, we note that in the paper we already include controlled experiments where the base model is fixed (Table 1 and Figure 3). Furthermore, as discussed above, we have conducted additional experiments using models from the Llama family, further confirming that TANGO’s effectiveness is not tied to any specific model architecture. We will make these points more explicit in the revised paper.
>
> > ***there are no guarantees for the convergence of the joint training procedure…I think a principle study could make the method even stronger.***
>
> Thank you for raising this insightful point. Theoretical convergence guarantees for co-evolving agents, such as our generator and verifier, remain an open challenge, especially in large-scale RL with LLMs.
>
> That said, we observe stable and consistent training across diverse settings. As shown in Figure 1 and Section 4.1, TANGO improves both training efficiency and final performance. Our ablation in Section 4.3 further shows that co-evolving both components yields significantly better results than fixing either one, supporting the stability of our interleaved optimization.
>
> Our setup resembles multi-agent RL (MARL) and two-timescale optimization, where both agents use policy gradients and mutually influence each other’s reward landscape. Theoretical tools from [6] may serve as a foundation to analyze TANGO’s convergence with suitable extensions for text-based feedback. While a complete convergence analysis remains beyond the scope of this work, we leave this as a promising direction for future work.
>
> > ***Comparison with Prover-Verifier Game [1,2] and EM-style training [3].***
>
> We thank the reviewer for the inspiring question. Compared to the **Prover-Verifier Game** [1,2], our approach differs in **two key aspects** among a few others:
>
> - Their verifier is trained using a supervised classification loss based on sneaky and helpful examples, whereas our verifier is optimized through RL;
> - Their verifier outputs a single scalar score that contributes to the outcome-level reward used in PPO training of the prover. In contrast, our verifier generates natural language step-level judgments, enabling fine-grained process-level rewards that more effectively guide the generator’s training.
>
> Regarding the second question, it is intriguing to draw connections between our proposed TANGO framework and **EM-style training** as explored in [3]. In that work, the LLM reasoning problem is formulated as sampling from the posterior $p(Z \mid X, Y)$, where $Z$ denotes the reasoning chain, $X$ the prompt, and $Y$ the outcome. Because this posterior is intractable, it is approximated via a GFlowNet in the E-step [3]. At a high level, we acknowledge the conceptual similarity noted by the reviewer. However, there are **two fundamental technical differences** that distinguish the two approaches:
>
> - While both involve an LLM-based generator, [3] introduces a GFlowNet specifically to approximate $p(Z \mid X, Y)$, a unique modeling choice. In contrast, TANGO incorporates a text-based generative verifier trained to evaluate the generator’s outputs, rather than to sample reasoning chains $Z$, thus serving a fundamentally different purpose;
> - The framework in [3] is grounded in the EM algorithm, optimizing expected log-likelihood. TANGO, on the other hand, is based on RL, directly optimizing expected return under both outcome-level and process-level rewards.
>
> We thank the reviewer for pointing out these connections, and we will cite and discuss these relevant works in the revised paper.
>
> > ***Currently, the verifier and the generator both use different models. I am curious if you have tried setting them to be the same, or for example having both models be of different capabilities to understand what role that plays?***
>
> Thank you for the constructive feedback. In our main math experiments, we use Qwen2.5-Math-7B as the generator and Qwen2.5-7B as the verifier, primarily because the verifier requires a larger context window to accommodate both the input question and the generator’s full output. In contrast, for the algorithmic reasoning experiments and the Llama experiments discussed above, we use Qwen2.5-1.5B or Llama-3.1-8B-Instruct for both the generator and verifier, thereby exploring the scenario where both components share the same base model. This design allows us to evaluate TANGO’s effectiveness under both settings.
>
> Generally speaking, there is no strict requirement on whether the generator and verifier should share the same base model or not. However, several considerations can guide the choice: the generator should possess reasonably good reasoning capabilities to start from, while the verifier should have a sufficiently large context window to process the full generator response for accurate step-level evaluation. Additionally, to ensure a fair comparison, we typically choose the verifier such that its reasoning ability is *not stronger* than that of the generator, in order to prevent potential “distillation” effects from the verifier to the generator. We will also make this point clearer in the revised paper.
>
> > ***For this setup, I believe an async training setup (e.g. [4,5]) could be quite useful to speed up training.***
>
> Thank you for your valuable suggestion. We agree with the reviewer that asynchronous training is a promising direction for improving training efficiency. We plan to explore this in future work to further speed up the training process of TANGO. In the revised paper, we will also explicitly discuss the potential of using asyn training and cite [4,5] for context.
>
> > ***The text explanation from the verifier is not used at all. Maybe it can be used for inference time search?***
>
> Thank you for your thoughtful suggestion. Yes, the generative process-level verifier (PRMs) trained by TANGO is indeed **well-suited** for inference-time scaling. Specifically, it can assign process-level scores to generator outputs, enabling inference-time search strategies such as Best-of-N selection and tree-based search. Moreover, unlike discriminative reward models, the generative verifier uniquely supports simultaneous scaling of both generator and verifier compute. By sampling multiple verification trajectories, we can improve verification accuracy, thereby providing more reliable supervision to the generator. That said, as this work primarily focuses on RL post-training, we reserve this exploration as future work.
>
> ---
>
> [1] Learning to Give Checkable Answers with Prover-Verifier Games. Anil et al. Arxiv 2021.
>
> [2] Prover-Verifier Games improve legibility of LLM outputs. Kirchner et al. Arxiv 2024.
>
> [3] Amortizing intractable inference in large language models. Hu et al. Arxiv 2023.
>
> [4] Asynchronous RLHF: Faster and More Efficient Off-Policy RL for Language Models. Noukhovitch et al. Arxiv 2024.
>
> [5] Trajectory Balance with Asynchrony: Decoupling Exploration and Learning for Fast, Scalable LLM Post-Training. Bartoldson et al. Arxiv 2025.
>
> [6] Multi-agent reinforcement learning: A selective overview of theories and algorithms. Zhang et al. Handbook of reinforcement learning and control (2021).
>
> ---
>
> We hope that our response has addressed all of your concerns, and that you may consider a favorable increase of the score. Please do not hesitate to discuss with us if you have any other comments.

---

> > ### Comment · Reviewer_esdQ · 2025-08-02
> >
> > Thanks for the thorough response!
> >
> > > we conducted additional experiments using Llama-3.1-8B-Instruct as the base model
> >
> > The Llama results are great, but it would be helpful to also include the base Llama3.1 (without SFT) results.
> >
> > > Compared to the Prover-Verifier Game [1,2], our approach differs in two key aspects
> >
> > I appreciate the author's elaboration on the differences. Perhaps I should have been more clear in my review. My purpose of pointing the connection to PVGs was to see if this framework can potentially help in similar tasks.
> >
> > > The framework in [3] is grounded in the EM algorithm, optimizing expected log-likelihood.
> >
> > [3] is pretty much just doing RL (it is just the PCL objective instead of REINFORCE/GRPO) if you consider just the E-step as the authors do in most of the experiments, so I think the approach is fairly close. Also similar to above, the comment was meant to spark perhaps ways in which the proposed TANGO approach can be incorporated in this setup!
> >
> > Overall, I think the response addressed my concerns so I will raise my score.

---

> ### Author Response · Authors · 2025-08-03
>
> Thank you very much for your feedback! We are glad to hear that our response addressed your concerns and appreciate your decision to raise your score. Below, we provide further response to your comments.
>
> > *It would be helpful to also include the base Llama3.1 (without SFT) results.*
>
> Thank you for your suggestion. Below we have updated the table to include the results of the base Llama-3.1-8B-Instruct model.
>
> | Model            | MATH500 | AIME2024 | AIME2025 | AMC2023 | OlympiadBench | Avg. |
> |:-----------------|:-------:|:--------:|:--------:|:-------:|:--------------:|:----:|
> | Llama3.1-8B-Instruct (Base)     |  46.4   |   3.3    |   3.3    |  22.5   |     13.6       | 17.9 |
> | TANGO-Llama3.1-8B-SFT     |  49.4   |   3.3    |   3.3    |  25.0   |     15.0       | 19.2 |
> | GRPO             |  56.2   |  10.0    |  3.3    |  35.0   |     20.9       | 25.1 |
> | **GRPO w/ TANGO**| **60.5**| **13.3** | **6.7** | **40.0**|   **23.6**     |**28.8**|
>
> Please note that the base model results shown here differ slightly from those in Table 2. This is because in Table 2, we report the higher numbers directly taken from prior work for system-level comparison, whereas here we report the results from our own evaluation for controlled ablation experiments.
>
> > *My purpose of pointing the connection to PVGs was to see if this framework can potentially help in similar tasks.*
>
> > *[3] is pretty much just doing RL (it is just the PCL objective instead of REINFORCE/GRPO) if you consider just the E-step as the authors do in most of the experiments, so I think the approach is fairly close. Also similar to above, the comment was meant to spark perhaps ways in which the proposed TANGO approach can be incorporated in this setup.*
>
> Thank you very much for the clarification. It helped us better understand the connection, and we now recognize that the combination of PCL and GFlowNet used in the E-step of [3] is conceptually similar to the use of GRPO with our generator during the generator update step in TANGO. We also agree that the core ideas behind our framework could inspire applications to similar tasks. A promising direction for future work is to also consider incorporating an RL-trained generative verifier’s process-level feedback with natural language reasoning into the M-step of [3], which could enable a more unified and interactive optimization process. We will clarify this connection and include a discussion in the updated paper.
>
> ---
>
> Once again, we thank the reviewer for your time and effort, and please do not hesitate to let us know if you have any further questions or comments about the paper.

---

### Official Review · Reviewer_5Vry · 2025-07-02

**Clarity:** 1
**Significance:** 3
**Originality:** 2
**Rating:** 5
**Confidence:** 5

**Summary:**

This paper introduces Tango, a reinforcement learning framework that jointly trains an LLM generator and verifier through interleaved RL updates. Unlike existing approaches that use fixed or discriminatively-trained verifiers, Tango employs a generative process-level verifier that is trained via RL alongside the generator. The key innovation is that the generative verifier learns to provide step-level feedback using only outcome-level supervision, eliminating the need for expensive step-level annotations. The framework creates a co-evolutionary dynamic where the generator produces increasingly diverse reasoning trajectories while the verifier learns to provide more accurate step-level judgments, leading to mutual reinforcement. Experiments on mathematical reasoning benchmarks demonstrate that both components achieve state-of-the-art performance among 7B/8B-scale models, with particularly strong improvements on challenging competition-level problems.

**Questions:**

Is there any reason to derive the final advantage as in (6)? Or is there a principle way to combine the advantage of different granularity?

**Ethical Concerns:**

["NO or VERY MINOR ethics concerns only"]

**Final Justification:**

The authors' clarification solves my concerns.

**Limitations:**

Though the evaluation includes out-of-domain tests, the training is still Qwen on math, which has been proven to be noisy as we can not disentangle whether the effectiveness comes from the base model or the algorithm itself [1,2].

[1] Spurious Rewards: Rethinking Training Signals in RLVR. Shao et al. 2025.

[2] The Unreasonable Effectiveness of Entropy Minimization in LLM Reasoning. Agarwal et al. 2025.

**Quality:**

3

**Strengths And Weaknesses:**

- It may not be intentional but it reads like this paper is the first to propose “a novel framework that uses RL to concurrently train both an LLM generator and a verifier in an interleaved manner”, which is incorrect and an overclaim if it did not explicitly say “generative verifier”. As mentioned in the introduction, PRIME also uses RL to concurrently train both an LLM generator and a verifier in an interleaved manner, though it uses a deterministic verifier, which, in my opinion, is the biggest distinction between Tango and PRIME.
- The method is straightforward, and the contribution mainly comes from engineering rather than novelty (I don’t mean it’s negative). As stated above, the biggest difference between Tangle and PRIME, the major baseline in this paper, is the adoption of generative verifier. However, the generative reward model is not new and has gained much attention to enhance verifier capability [1, 2]. Therefore, the method to replace the determinative verifier in PRIME’s framework with a generative one is straightforward. Nevertheless, making the recipe work still requires dedicated efforts, and the conclusion that generative verifiers work in training may still inspire future work along with improving supervision in RL, so this paper is meaningful in this sense.
- The results show solid improvements over baselines and generalize outside math, and the ablation study validates (1) how the supervision signal improves along the co-evolution with the policy model, and (2) how the policy model benefits from mitigating reward hacking. Such results are meaningful in provoking a second thought on recent fever over RL with verifiable rewards.

[1] Generative Reward Models. Mahan et al. 2024.

[2] Process Reward Models That Think. Khalifa et al. 2025.

---

> ### Author Rebuttal · Authors · 2025-07-31
>
> Dear Reviewer 5Vry,
>
> Thank you for acknowledging the contributions of our work, particularly the significance of incorporating a generative verifier within a co-evolutionary training framework to advance RL supervision, the dedicated efforts required to make this recipe work in practice, the strong empirical performance and generalization beyond math, and the detailed and meaningful ablation studies. In the following, we address your concerns in detail.
>
> > ***It may not be intentional but it reads like this paper is the first to propose “a novel framework that uses RL to concurrently train both an LLM generator and a verifier in an interleaved manner” ... PRIME also uses RL to concurrently train both an LLM generator and a verifier in an interleaved manner, though it uses a deterministic verifier...***
>
> > ***... the major baseline in this paper, is the adoption of generative verifier. However, the generative reward model is not new and has gained much attention to enhance verifier capability [1, 2]...***
>
> Thank you for pointing this out. We would like to clarify that the verifier in PRIME is a logit-based model trained via **supervised fine-tuning (SFT)** with a cross-entropy loss, rather than RL. In contrast, our framework uses RL to concurrently train both the generator and the verifier in an interleaved manner. The generative verifiers (reward models) in [1, 2] are also trained via **SFT**. Therefore, we believe it is accurate to state that our work is the first to propose a framework that uses RL to concurrently train both an LLM generator and a (generative) verifier in an interleaved manner. We will also make this point clearer in the revised paper.
>
> We would also like to emphasize that this concurrent RL training of both components is not merely a swap-in replacement, but **central** to the success of our co-evolving framework. The effectiveness of such a system depends on both the generator and verifier being sufficiently capable and improving in tandem. If one component significantly lags behind (for example, if the verifier is trained only with SFT, which is known to generalize poorly in complex reasoning tasks [3]), it can disrupt the learning dynamics and hinder mutual progress. By contrast, RL enables both components to continually adapt to each other's evolving behaviors, which is essential for robust generalization and effective reasoning. That’s also the key reason why our framework outperforms PRIME.
>
> That said, we are happy to revise “verifier” to “generative verifier” for clarity and precision, and we remain open to further rewording to avoid any potential ambiguity or overstatement, should any concerns persist.
>
> > ***The method is straightforward, and the contribution mainly comes from engineering rather than novelty (I don’t mean it’s negative) ... Nevertheless, making the recipe work still requires dedicated efforts…***
>
> We are very delighted and grateful to see that the reviewer recognized and appreciated our contributions. However, we believe that novelty in deep learning research, especially in the LLM era, can manifest in various ways. Often, the most impactful advances arise not from introducing entirely new components, but from developing approaches that are particularly **well-suited to a specific context**, thereby enabling a major performance boost. Similarly, the novelty of our paper stems from recognizing that jointly training the verifier and generator is essential, and that such training *cannot* be effectively achieved with a verifier trained via SFT. Instead, RL is required for both components. (Our novelty is also acknowledged by Reviewer esdQ.) This design, while conceptually **simple**, is nontrivial to realize and proves **highly effective** in practice, leading to a substantial boost in performance.
>
> > ***Is there any reason to derive the final advantage as in (6)? Or is there a principle way to combine the advantage of different granularity?***
>
> Thank you for the insightful question. Intuitively, Eq. (6) computes the advantage at the at the $t$-th token as the sum of normalized step rewards whose end-token indices (denoted by $I(k)$) are no smaller than $t$. This design aligns with the the process-level advantage computation used in GRPO by DeepSeek [4] (see Section 4.1.3).
>
> Our approach to combining advantages at different granularities, step-level and outcome-level, is defined in Eq. (7). We believe this provides a **principled** way to integrate multi-granularity advantage signals and has the potential for broader applicability.
>
> Specifically, we first compute step-level and outcome-level advantages independently and then combine them using Eq. (7) with a scheduling coefficient $\alpha$. We highlight two key design choices that are crucial to the success of our method:
>
> - We apply an exponential decay schedule to $\alpha$, which is essential to TANGO’s success. Early in training, step-level supervision has a stronger influence to encourage exploration of reasoning strategies. As training progresses, we gradually reduce its weight to promote stable convergence and mitigate reward hacking.
> - Empirically, we find that computing and normalizing the step and outcome advantages separately before combining them yields significantly more stable learning than merging the rewards first and computing a single advantage. This is because advantage normalization depends on the scale and distribution of the underlying rewards. Merging step and outcome rewards before normalization could distort their relative contributions due to scale mismatch, resulting in instability and degraded performance. By normalizing each advantage independently, we preserve their intended effects prior to aggregation.
>
> We will expand on the discussion above and include it in the revised paper.
>
> > ***the training is still Qwen on math, which has been proven to be noisy as we can not disentangle whether the effectiveness comes from the base model or the algorithm itself.***
>
> We thank the reviewer for raising this important point. To better address the reviewer’s concern, we conducted additional experiments **using Llama-3.1-8B-Instruct as the base model**, following the same training/evaluation protocol as in Table 1. The results are presented in the table below.
>
> **Base Model: Llama-3.1-8B-Instruct**
> | Model            | MATH500 | AIME2024 | AIME2025 | AMC2023 | OlympiadBench | Avg. |
> |:-----------------|:-------:|:--------:|:--------:|:-------:|:--------------:|:----:|
> | TANGO-Llama3.1-8B-SFT     |  49.4   |   3.3    |   3.3    |  25.0   |     15.0       | 19.2 |
> | GRPO             |  56.2   |  10.0    |  3.3    |  35.0   |     20.9       | 25.1 |
> | **GRPO w/ TANGO**| **60.5**| **13.3** | **6.7** | **40.0**|   **23.6**     |**28.8**|
>
> They show that even when Llama is used as the base model, TANGO continues to deliver significant improvements, demonstrating strong generalization across different model families. This further confirms that TANGO’s effectiveness **stems from** our interleaved RL co-evolving framework, rather than being specific to any particular model architecture. We will include the above results and discussion in the revised paper.
>
> ---
>
> [1] Generative Reward Models. Mahan et al. Arxiv 2024.
>
> [2] Process Reward Models That Think. Khalifa et al. Arxiv 2025.
>
> [3] SFT Memorizes, RL Generalizes: A Comparative Study of Foundation Model Post-training. Chu et al. ICML 2025.
>
> [4] DeepSeekMath: Pushing the Limits of Mathematical Reasoning in Open Language Models. Shao et al. Arxiv 2024.
>
> ---
>
> Thank you again for your time and feedback. We hope that our response has adequately answered your questions, and would lead to a favorable increase of the score. We are happy to discuss more if you have any further questions.

---

> > ### Comment · Reviewer_5Vry · 2025-08-02
> >
> > Thank you for the clarification, and it solves my concerns.

---

> > > ### Author Response · Authors · 2025-08-03
> > >
> > > Thank you very much for your feedback! We are glad to hear that our response addressed your concerns. Please do not hesitate to let us know if you have any further questions or comments about the paper.

---

### Official Review · Reviewer_A2Yu · 2025-07-03

**Clarity:** 3
**Significance:** 3
**Originality:** 3
**Rating:** 5
**Confidence:** 4

**Summary:**

In this paper, the authors address the limitations of vanilla fixed verifiers which are prone to reward hacking and exhibit poor generalization. To overcome these issues, they propose a framework called TANGO, which employs reinforcement learning to jointly train an LLM generator and a verifier in an interleaved manner. Notably, TANGO dynamically trains a process-level LLM verifier using only outcome-level rewards, thereby improving training efficiency through more fine-grained reward signals.

**Questions:**

Please see the weakness.

**Ethical Concerns:**

["NO or VERY MINOR ethics concerns only"]

**Final Justification:**

The response has solved my concerns.

- The generalizability of the proposed method has been verified on other LLM backbones.
- The model is more efficient with smaller training steps.
- The ablation study is sufficient.

Thus I raise the score.

**Limitations:**

Please see the weakness.

**Paper Formatting Concerns:**

None.

**Quality:**

3

**Strengths And Weaknesses:**

Strengths:
1. The authors train a verifier that combines the advantages of both outcome-level and process-level supervision, enriching the reward signals available during training.
2. The proposed interleaved RL training framework enables the generator and verifier to be jointly optimized, allowing them to continuously improve in a co-evolutionary manner.

Weaknesses:
1. The experiments are conducted solely on the Qwen series of models, which is insufficient to demonstrate the generalizability of the proposed method. Including results on additional base models would make the findings more convincing.
2. The efficiency is a concern. Since TANGO requires interleaved training between the generator and verifier, it introduces higher computational costs. For instance, as shown in Figure 1, the verifier undergoes a 40-step warm-up and must be periodically updated during training. The authors should ensure that vanilla RL baselines are granted equivalent training steps or report the actual wall-clock time to reflect computational efficiency.
3. The ablation study is insufficient. A dedicated analysis of the process-level reward is necessary to validate the effectiveness of the implicit step-wise supervision.
4. It could be better to outline the contributions in the Introduction section to improve the overall readability and clarity of the paper. The code has not been released, which limits reproducibility and reduces the potential impact of this work on the broader research community.

---

> ### Author Rebuttal · Authors · 2025-07-31
>
> Dear Reviewer A2Yu,
>
> Thanks for your insightful and detailed feedback. We appreciate your recognition of TANGO’s strengths and contributions, particularly its enriched reward signals and co-evolutionary training framework. Below we address your concerns one by one.
>
> > ***Including results on additional base models would make the findings more convincing.***
>
> Thank you for the valuable suggestion. We agree with the reviewer that including results on additional base models would strengthen the validity of our findings. Following the reviewer’s suggestion, we conducted experiments **using Llama-3.1-8B-Instruct as the base model** for both the generator and verifier, following the same training and evaluation protocol as in Table 1. The results are presented in the table below.
>
> **Base Model: Llama-3.1-8B-Instruct**
> | Model            | MATH500 | AIME2024 | AIME2025 | AMC2023 | OlympiadBench | Avg. |
> |:-----------------|:-------:|:--------:|:--------:|:-------:|:--------------:|:----:|
> | TANGO-Llama3.1-8B-SFT     |  49.4   |   3.3    |   3.3    |  25.0   |     15.0       | 19.2 |
> | GRPO             |  56.2   |  10.0    |  3.3    |  35.0   |     20.9       | 25.1 |
> | **GRPO w/ TANGO**| **60.5**| **13.3** | **6.7** | **40.0**|   **23.6**     |**28.8**|
>
> These results show that even when using Llama as the base model, TANGO continues to deliver significant improvements, demonstrating strong generalization across different model families. This further confirms that TANGO’s effectiveness **stems from** our interleaved RL co-evolving framework, rather than being specific to any particular model family. We will include the above results and discussion in the revised paper.
>
> > ***The efficiency is a concern … The authors should ensure that vanilla RL baselines are granted equivalent training steps ... to reflect computational efficiency.***
>
> Thank you for the question. We would like to clarify that for Figures 1 and 2, as well as Table 1, we followed the same evaluation protocol as the PRIME [1] paper, comparing our method to vanilla RL baselines based on the number of generator update steps. However, we agree with the reviewer that from an efficiency standpoint, it is important to ensure that vanilla RL baselines are granted equivalent total training steps. In response to the reviewer’s suggestion, we extended the experiments in Figure 3 to include detailed step counts and additional comparisons with all baselines (vanilla RL, ORM, PRIME). The results are presented in the table below.
> | Model            | G training steps | V training steps | Total training steps | MATH500 | AIME2024 | AIME2025 | AMC2023 | OlympiadBench | Avg. |
> |:-----------------|:-------:|:-------:|:----------------:|:-------:|:--------:|:--------:|:-------:|:--------------:|:----:|
> | GRPO (Original)             |   200     |   0     |        200         |  74.6   |   13.3   |   10.0   |  50.0   |     36.9       | 37.0 |
> | GRPO  (Extended)           |   400     |   0     |        400         |  76.8   |   13.3   |   13.3   |  55.0   |     38.8       | 39.4 |
> | GRPO + ORM       |   200     |   200 (pretrained)     |        400         |  75.8   |   13.3   |   16.7   |  55.0   |     39.1       | 40.0 |
> | PRIME (GRPO + PRM)       |   200     |   200 (alternating)     |        400         |  79.4   |   16.7   |   13.3   |  60.0   |     41.9       | 42.3 |
> | **GRPO w/ TANGO**|   200     |   40 (warmup) + 66 (alternating)     |        **306**         | **81.4**| **20.0** | **20.0** | **65.0**|   **43.9**     |**46.1**|
>
> Specifically, we added an experiment that extends vanilla GRPO training from 200 to 400 steps. As shown in the table, doubling the training steps yields limited performance gains, particularly on the most challenging benchmarks such as AIME2024 and AIME2025, where performance nearly plateaus after 200 steps. For the stronger baselines, GRPO + ORM and PRIME, both require training a verifier (reward model). In GRPO + ORM, the ORM is pretrained for 200 steps. In PRIME, the PRM is updated alternately with the generator in a 1:1 ratio, also resulting in 200 update steps. In our method, GRPO w/ TANGO, the verifier is first warmed up for 40 steps and then updated alternately with the generator at a 3(G):1(V) ratio, resulting in 66 additional verifier updates, yielding a total of 106 verifier update steps.
>
> Taken together, these results show that our method achieves the best performance while **using fewer training steps**. This highlights that TANGO’s improvements stem from our proposed training framework rather than from increased training steps. We will update the above results and discussion in the revised paper.
>
> > ***The ablation study is insufficient. A dedicated analysis of the process-level reward is necessary to validate the effectiveness of the implicit step-wise supervision.***
>
> We thank the reviewer for the suggestion. First, we would like to clarify that the results presented in Figure 3 of the main paper (which we have augmented and summarized in the table above) already ablate the effectiveness of our RL-trained step-wise supervision, in comparison to: (1) SFT-trained logit-based token-level supervision (PRIME), (2) outcome reward from a pretrained ORM (GRPO + ORM), and (3) rule-based outcome reward (vanilla GRPO).
>
> To further validate the effectiveness of our step-wise reward design, we conducted two additional ablations:
>
> 1. **Removing step normalization in Eq. (3)**: We change the reward from the normalized form $r_{g,\text{step}}^{I(k)}(o_g)= \frac{y_{\text{step},k}}{K}$ to the unnormalized form $r_{g,\text{step}}^{I(k)}(o_g)= y_{\text{step},k}$;
> 2. **Removing intermediate step-level supervision**: Instead of assigning step-wise rewards at each reasoning step, we aggregate the step signals into a single scalar $\frac{\sum_{k=1}^K |{y}_{\text{step},k}|}{K}$ and add it to the final outcome reward. This setting reduces step supervision to a coarse auxiliary signal at the final step.
>
> We summarized all the results in the table below.
>
> | Model            | Outcome Reward | Process Reward | Avg. accuracy on five math benchmarks |
> |:----------------- |:----------------:|:----------------:|:----:|
> | GRPO  (400 steps)           |   rule-based, hard        |  no process reward | 39.4 |
> | GRPO + ORM       |   soft from ORM  |  no process reward  | 40.0 |
> | GRPO w/ TANGO (removing step-level supervision) |  rule-based, hard + RL-trained soft step accuracy | no process reward | 42.0 |
> | PRIME (GRPO + PRM)       |   rule-based, hard | SFT-trained, token-level        |  42.3 |
> | GRPO w/ TANGO (removing step norm) |  rule-based, hard | RL-trained, step-level (no step norm) | 43.4 |
> | **GRPO w/ TANGO** |  rule-based, hard | RL-trained, step-level (w/ step norm) | **46.1** |
>
> The results above reveal several key findings:
>
> - Even when step-wise signals are only used as an auxiliary term in the final reward (without intermediate supervision), the method still outperforms the ORM baseline, validating the effectiveness of the learned step-level supervision;
> - However, assigning supervision explicitly at intermediate steps as process rewards is significantly more effective, as it enables more accurate credit assignment and facilitates early-stage exploration;
> - Normalizing step-wise rewards by the number of reasoning steps, as done in our method, is critical. It removes the policy's bias toward longer or shorter step lengths, allowing the generator to flexibly determine the appropriate number of steps for each problem.
>
> We will add the above results and discussion in the revised paper.
>
>  > ***It could be better to outline the contributions in the Introduction section to improve the overall readability and clarity of the paper.***
>
> Thanks for the constructive suggestion. We will include a clear list of contributions in the Introduction section in the revised paper.
>
>  > ***The code has not been released.***
>
> In fact, we have already released the code to ensure reproducibility. However, to preserve the integrity of the double-blind review process, we omitted the code link in the current submission. The code link will be included in the camera-ready version upon acceptance.
>
> ---
>
> [1] Process Reinforcement through Implicit Rewards. Cui et al. Arxiv 2025.
>
> ---
>
> We hope our response has thoroughly addressed your concerns, and would really appreciate it if you could consider raising your score accordingly. If you have any further questions or suggestions, please do not hesitate to share them. We are eager to engage in further discussions with you.

---

> > ### Comment · Reviewer_A2Yu · 2025-08-02
> >
> > Thank you for the clarification. I have raised my score as it solves my concerns.

---

> > > ### Author Response · Authors · 2025-08-03
> > >
> > > Thank you very much for your feedback! We are glad to hear that our response addressed your concerns and appreciate your decision to raise your score. Please do not hesitate to let us know if you have any further questions or comments about the paper.

---

### Official Review · Reviewer_DEZv · 2025-07-06

**Clarity:** 3
**Significance:** 3
**Originality:** 2
**Rating:** 5
**Confidence:** 3

**Summary:**

The paper proposes to jointly train a "generator" policy model along with a process reward model "verifier", using RL from the outcome-level rewards only. This leads to strong empirical gains in math and reasoning tasks at the 7/8B model scale.

**Questions:**

Q1. The claim the generative verification mitigates reward hacking is intriguing. Aside from the overall strong performance of the method, is there any more specific evidence that mitigating reward hacking is one of the mechanisms by which performance improves?

**Ethical Concerns:**

["NO or VERY MINOR ethics concerns only"]

**Final Justification:**

This paper proposes generative reward models as a more robust approach for process-level rewards. The empirical results are strong and the approach is conceptually well-motivated. I recommend accept.

**Limitations:**

Yes, although some of these math datasets are quite small so it would be good to have confidence intervals or some other measure of finite-sample variance

**Paper Formatting Concerns:**

no concerns

**Quality:**

3

**Strengths And Weaknesses:**

S1. The primary strength of the paper is empirical: the trained process verifier achieves substantial performance improvements, apparently over strong baselines.
S2. The proposed approach also yields text-based judgments, which the authors claim mitigates reward hacking.
S3. The experiments seem solid and rigorous, although these reasoning tasks are not my area of expertise.

W1. While the approach is well-motivated at a high level, there is relatively little conceptual novelty and a considerable amount of heuristic reward/advantage shaping (sec 3.2). However, I think that's okay for this type of paper so I am still recommending accept.

---

> ### Author Rebuttal · Authors · 2025-07-31
>
> Dear Reviewer DEZv,
>
> Thank you very much for your constructive comments and positive feedback. It is heartening to note that you found the proposed approach well-motivated, the experiments solid and rigorous, and the empirical results strong. Below, we address your questions one by one and provide additional clarifications.
>
> > ***W1. Clarification on the novelty of our paper.***
>
> We thank the reviewer for acknowledging and recognizing the contributions of our work. We believe that novelty in deep learning research, especially in the LLM era, can manifest in various ways. Often, the most impactful advances arise not from introducing entirely new components, but from developing approaches that are particularly **well-suited to a specific context**, thereby enabling a major performance boost. Similarly, the novelty of our paper stems from recognizing that jointly training the verifier and generator is essential, and that such training *cannot* be effectively achieved with a verifier trained via SFT. Instead, RL is required for both components. (Our novelty is also acknowledged by Reviewer esdQ.) This design, while conceptually **simple**, is nontrivial to realize and proves **highly effective** in practice, leading to a substantial boost in performance.
>
> > ***Q1. ... is there any more specific evidence that mitigating reward hacking is one of the mechanisms by which performance improves.***
>
> Thanks for the question. As the reviewer noted, the strong empirical results support this claim. Below, we provide both an intuitive explanation and concrete evidence for why we believe TANGO’s generative, text-based verifier mitigates reward hacking and contributes to improved performance:
>
> Compared to a logit-based process verifier, a generative, text-based verifier is inherently more robust against exploitation. Consider how a generator might attempt to maximize process rewards in each case. With a logit-based verifier, the generator can game the reward by producing sequences that merely increase token-level log probabilities under the verifier model, regardless of whether the underlying reasoning is actually correct. In contrast, a text-based verifier evaluates reasoning by generating natural language judgments (e.g., stating whether a step is valid and why), which requires the generator to produce semantically coherent and logically sound reasoning in order to obtain a high reward. This makes shallow optimization strategies far less effective and substantially reduces the risk of reward hacking.
>
> The ablation results shown in Figure 3 of the main paper further support this intuition: compared to PRIME, which uses a logit-based PRM, TANGO, with its generative process-level verifier, achieves better performance, even though both approaches involve co-training a generator and a verifier.
>
> We will add the above discussion in the revised paper.
>
> > ***It would be good to have confidence intervals or some other measure of finite-sample variance.***
>
> Thank you for the valuable suggestion. We agree that reporting error bars from multiple independent runs is especially important for smaller math datasets such as AIME24, AIME25, and AMC23. Due to limited time and computational resources during the rebuttal period, and the high cost of RL with 7B-scale models, we report here the mean and standard deviation of accuracies for GRPO and GRPO w/ TANGO (corresponding to the second and third rows of Table 1 in the main paper) based on three independent runs across the five math datasets, as shown in the table below. These results demonstrate a relatively **stable** trend across runs. We will include error bars for the remaining results in the revised paper.
>
> | Model            | MATH500 | AIME2024 | AIME2025 | AMC2023 | OlympiadBench |
> |:-----------------|:-------:|:--------:|:--------:|:-------:|:--------------:|
> | GRPO             |  74.4 ± 0.3  |   11.1 ± 1.6   |   8.9 ± 1.6   |  50.8 ± 1.2   |     37.1 ± 0.3       |
> | **GRPO w/ TANGO**| **81.3 ± 0.2**| **18.9 ± 1.6** | **18.9 ± 4.2** | **62.5 ± 2.0**|   **43.5 ± 1.1**     |
>
> ---
>
> We hope our response has addressed all of your concerns and can lead to a favorable increase of the score. Please feel free to let us know if you have other questions or suggestions.

---

> > ### Comment · Reviewer_DEZv · 2025-08-05
> >
> > Thanks for the reply and for the confidence interval calculations. They reinforce my view that the paper should be accepted.

---

> > > ### Author Response · Authors · 2025-08-05
> > >
> > > Thank you very much for your feedback and positive response! We truly appreciate your thoughtful comments and suggestions.

---

### Decision · Program_Chairs · 2025-09-17

**Decision:**

Accept (poster)

**Comment:**

This paper studies joint RL training of generator and verifier LLMs in reasoning tasks, i.e., ones where a latent token sequence intervenes in the generation of the answer given the query. The generator receives trajectory-level rewards for the correctness of its answer and step-level rewards from the verifier. The verifier, which also uses latent reasoning, receives a reward for correctly classifying the generator's reasoning as correct or incorrect. Various techniques are proposed to enable the joint optimisation of the two models. Evaluations on a variety of tasks (mainly maths benchmarks) show a consistent improvement over baselines.

The main reasons to accept, all mentioned by multiple reviewers, are (1) the substantial performance improvements given by the proposed algorithm, (2) the interesting effect of the verifier's latent reasoning promoting explainability and mitigating reward hacking, (3) the diversity of tasks considered and the demonstration of generalisation abilities outside of maths tasks.

The authors' questions were almost entirely addressed in the rebuttal. Several reviewers note that the main contributions are empirical and the conceptual novelty is not very high. However, the reviewers unanimously give a rating of Accept, and I follow their recommendation. The authors are encouraged to take account of the reviewers' suggestions when preparing the final version.